# A Scalable Solver for 2p0s Differential Games with One-Sided Payoff Information and Continuous Actions, States, and Time

## Abstract

Existing solvers for imperfect-information extensive-form games (IIEFGs) often struggle with scalability in terms of action and state space sizes and the number of time steps. However, many real-world games involve continuous action and state spaces and occur in continuous time, making them differential in nature. This paper addresses the scalability challenges for a representative class of two-player zero-sum (2p0s) differential games where the informed player knows the game type (payoff) while the uninformed one only has a prior belief over the set of possible types. Such games encompass a wide range of attack-defense scenarios, where the defender adapts based on their belief about the attacker's target. We make the following contributions: (1) We show that under the Isaacs' condition, the complexity of computing the Nash equilibrium for these games is not related to the action space size; and (2) we propose a multigrid approach to effectively reduce the cost of these games when many time steps are involved. Code for this work is available at anonymous repo.

## 1. Introduction

The strength of game solvers has grown rapidly in the last decade, beating elite-level human players in Chess (Silver et al., 2017a), Go (Silver et al., 2017b), Poker (Brown & Sandholm, 2019; Brown et al., 2020b), Diplomacy (FAIR† et al., 2022), Stratego (Perolat et al., 2022), among others with increasing complexity. Most of the existing solvers with proved convergence, e.g., CFR+ variants (Tammelin, 2014; Burch et al., 2014; Moravčík et al., 2017; Brown et al., 2020b; Lanctot et al., 2009), FTRL variants (McMahan, 2011; Perolat et al., 2021), and mirror descent vari-

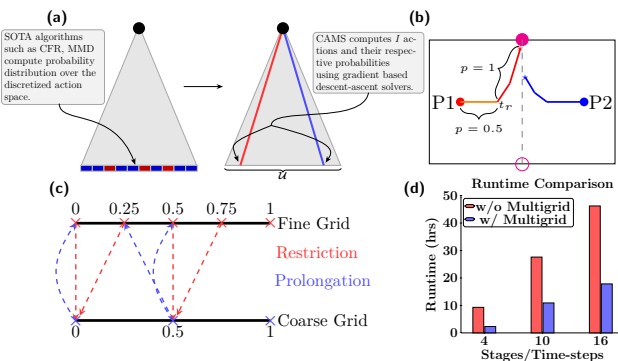

*Figure 1.* **(a)** We explain the atomic nature of equilibrium strategies in the games of our interest. Exploiting this nature allows us to tractably solve games with continuous action spaces. **(b)** Sample equilibrium state trajectories of a 2p0s differential game where P2 guesses P1's target (magenta circles). P1's optimal strategy is to reveal his target after a critical time $t_r$. **(c)** Illustration of a 2-level multigrid solver. Fine grid errors are restricted to the coarse grid, where cheap corrections are computed and prolongated to the fine grid. **(d)** Multigrid further accelerates value approximation for games with various number of time steps.

ants (Sokota et al., 2022; Cen et al., 2021; Vieillard et al., 2020), are designed for games with finite action and state sets, and have computational complexities increasing along the sizes of these sets. Real-world imperfect-information games, however, can often have continuous action and state spaces and happen in continuous time, making them differential in nature. Directly applying the existing solvers to these differential games would require either insightful action-state-time abstraction or enormous compute. Neither are readily available.

This paper addresses this scalability challenge for a representative subset of 2p0s differential games where the informed player knows the game type (payoff) while the uniformed player only has a prior belief $p_0 \in \Delta(I)$ over a set of $I$ possible types. We also assume that the Isaacs' condition holds, i.e., the complete-information version of the game has a pure Nash equilibrium. This condition commonly holds for differential games with control-affine dynamics. While restricted, such games represent a wide range of attack-defense scenarios that can be described as follows: At the beginning of the game, nature draws a game type according to $p_0$ and informs the informed player (P1) about the type.

[1]Anonymous Institution, Anonymous City, Anonymous Region, Anonymous Country. Correspondence to: Anonymous Author <anon.email@domain.com>.

Preliminary work. Under review by the International Conference on Machine Learning (ICML). Do not distribute.

As the game progresses, the belief about the true game type, which is assumed to be public knowledge, is updated from $p_0$ based on the action sequence taken by P1 and his public strategy profile via the Bayes' rule. P1's (resp. P2's) goal is to minimize (resp. maximize) the expected payoff over $p_0$. Due to the zero-sum nature, P1 may need to delay information release or manipulate P2's belief to take full advantage of information asymmetry; and P2's strategy is to minimize a worst-case risk. Some real-world examples of the game include football set-pieces where the attacker has private information about which play is to be executed, and missile defense where multiple potential targets are concerned. The setting of one-sided information, i.e., P1 knows everything about P2, is necessary for P2 to derive defense strategies in risk-sensitive games.

We claim the following contributions:

- We explain that the computational complexity for approximating the Nash equilibrium for the games of our interest is related to the number of game types ($I$) rather than to the action space size.
- We explain that the equilibrium values for the informed and uninformed players can be computed via two separate backward induction processes through a primal-dual formulation of the game.
- We propose a multigrid approach to tractably solve games with continuous state spaces and many time steps. Empirical results show that our solver outperforms SOTA IIEFG solvers including CFR+ (Tammelin, 2014), MMD (Sokota et al., 2022), Deep-CFR (Brown et al., 2019), and a SOTA continuous-action solver JPSPG (Martin & Sandholm, 2024), on games of our interest. Our solver also approximates reasonable strategies for game settings that are intractable for SOTA solvers.

## 2. Related Work

**2p0s games with incomplete information.** Games where players have missing information only about the game types are often called *incomplete-information* games. These games are a subset of imperfect-information games where nature plays a chance move at the beginning (Harsanyi, 1967). The seminal work of (Aumann et al., 1995) developed equilibrium strategies for a repeated and one-sided setting of such games through the "Cav u" theorem, which relates the value of the game with that of a *non-revealing* version of the game where both players only know the distribution of the game type. Briefly, the "Cav u" theorem reveals that belief-manipulating behavioral strategies are necessary to achieve value convexification and thus the equilibrium. As we will discuss, this theorem plays an important role in enabling scalable solve of games with continuous action spaces. Building on top of (Aumann et al., 1995), (De Meyer, 1996) introduced a dual game in which the behavioral strategy of the uninformed player becomes Markov.

This technique helped (Cardaliaguet, 2007; Ghimire et al., 2024) to establish the value existence proof for 2p0s differential games with incomplete information with and without state constraints. Unlike repeated games where belief manipulation occurs only in the first round of the game, differential games may have multiple critical time-state-belief points where belief manipulation is required to achieve equilibrium, depending on the specifications of system dynamics, payoffs, and state constraints (Ghimire et al., 2024).

**IIEFGs.** IIEFGs represent the more general set of multi-agent decision-making problems with finite horizons. Since any 2p0s IIEFG with finite action sets has a normal-form formulation, a unique Nash equilibrium always exists in the space of mixed strategies. Significant efforts have been taken to approximate equilibrium of large IIEFGs (Koller & Megiddo, 1992; Billings et al., 2003; Gilpin & Sandholm, 2006; Gilpin et al., 2007; Sandholm, 2010; Brown & Sandholm, 2019) leading to algorithms that are no-regret and with sublinear or linear convergence rates (Zinkevich et al., 2007; Abernethy et al., 2011; McMahan, 2011; Tammelin, 2014; Johanson et al., 2012; Lanctot et al., 2009; Brown et al., 2019; 2020a; Perolat et al., 2021; Sokota et al., 2022; Perolat et al., 2022; Schmid et al., 2023) (see summary in Tab. 1). Notably, these algorithms have computational complexities increasing with the action space size $U$, provided that the equilibrium behavioral strategy lies in the interior of the simplex $\Delta(U)$ (see discussion in Appendix F). Critically, this assumption does not hold for differential games equipped with the Isaacs' condition, in which case the equilibrium strategy is mostly pure along the game tree, and is atomic on the action space $\mathcal{U}$ when mixed, as we explain in Sec. 4. While studies on continuous action normal- and extensive-form games exist (Martin & Sandholm, 2024; 2023), these methods are restricted to a class of games that either admit a pseudoconcave potential or are monotone.

*Table 1.* Solver computational complexity (best case) with respect to action space $\mathcal{A}$ and equilibrium error $\varepsilon$

| Algorithm | Complexity |
|---|---|
| CFR variants (Zinkevich et al., 2007; Lanctot et al., 2009; Brown et al., 2019; Tammelin, 2014; Johanson et al., 2012) | $\mathcal{O}(U\varepsilon^{-2})$ to $\varepsilon$-Nash |
| FTRL variants & MMD (McMahan, 2011; Perolat et al., 2021; Sokota et al., 2022) | $\mathcal{O}\left(\frac{U}{\varepsilon}\ln\left(\frac{1}{\varepsilon}\right)\right)$ to $\varepsilon$-QRE |

**Descent-ascent algorithms for nonconvex-nonconcave minimax problems.** Existing developments in IIEFGs focused on convex-concave minimax problems due to the bilinear form of the expected payoff through the conversion of games to their normal forms. This paper, on the other hand, investigates the nonconvex-nonconcave minimax problems to be solved at every infostate when actions are considered continuous. To this end, we use the doubly smoothed gradient descent ascent method (DS-GDA) which

has a worst-case complexity of $\mathcal{O}(\varepsilon^{-4})$ (Zheng et al., 2023).

**Multigrid methods for accelerating value approximation.** Multigrid methods (Trottenberg et al., 2000) are widely used to accelerate PDE solving on a mesh (e.g., fluid mechanics). In a typical V-cycle (Braess & Hackbusch, 1983), a few iterations of relaxation (e.g., Gauss-Seidel) are first performed on a fine mesh, and the resulting residual is restricted to a coarser mesh, where a PDE correction is solved and prolongated to the fine mesh. Essentially, the V-cycle uses a coarse solve to reduce the low-frequency approximation error in the PDE solution at a low cost, leaving only the high-frequency errors to be resolved through the fine mesh and resulting in faster solution convergence than conventional PDE solvers. Multigrid has been successfully applied to solving Hamilton-Jacobi-Bellman (HJB) and Hamilton-Jacobi-Isaacs (HJI) equations (Han & Wan, 2013) for optimal control problems and differential games. Nonetheless, extending multigrid to incomplete-information differential games and value approximation based on neural nets has rarely been discussed.

## 3. Problem Statement

**Notations and preliminaries.** We denote by $\Delta(I)$ the simplex in $\mathbb{R}^I$, $[T] := \{1, ..., T\}$, $a[i]$ the $i$th element of vector $a$, $\partial_p V$ the subgradient of function $V$ with respect to $p$. Consider a time-invariant dynamical system that defines the evolution of the joint state $x \in \mathcal{X} \subseteq \mathbb{R}^{d_x}$ of P1 and P2 with control inputs $u \in \mathcal{U}$ and $v \in \mathcal{V}$, respectively:

$$\dot{x}(t) = f(x(t), u, v). \tag{1}$$

The game starts at $t_0 \in [0, T]$ from some initial state $x(t_0) = x_0$. The initial belief $p_0 \in \Delta(I)$ is set to nature's distribution about the game type. P1 of type $i$ accumulates a running cost $l_i(u, v)$ during the game and receives a terminal cost $g_i(x(T))$. The goal of P1 is to minimize the expected sum of the running and terminal costs, which P2 maximizes.

Denote by $\{\mathcal{H}_r^i(t)\}^I$ the joint sets of behavioral strategies of P1, and $\mathcal{Z}_r(t)$ the set of behavioral strategies of P2. P1 chooses his strategy $\eta_i \in \mathcal{H}_r^i(t)$ according to his type $i$, while P2's strategy $\zeta \in \mathcal{Z}_r(t)$ is independent of $i$. At any game tree node $(t, x, p) \in [0, T] \times \mathcal{X} \times \Delta(I)$, $\eta_i$ (resp. $\zeta$) is a probability measure over $\mathcal{U}$ (resp. $\mathcal{V}$), and players move simultaneously. With mild abuse of notation, let $(\eta(t), \zeta(t))$ be the random open-loop controls $(\alpha_\omega(t), \delta_\omega(t))$ induced by $(\eta, \zeta)$ and determined by the random seed $\omega^1$. $X_{t_1}^{t_0, x_0, \eta_i, \zeta}$ is then the random state arrived at $t_1$ from $(t_0, x_0)$ following $(\eta_i, \zeta)$ and the system dynamics in Eq. 1. The loss of P1 in a type-$i$ game is:

$$J_i(t_0, x_0; \eta_i, \zeta) := g_i\left(X_T^{t_0, x_0, \eta_i, \zeta}\right) + \int_{t_0}^{T} l_i(\eta_i(s), \zeta(s))ds,$$

---

[1]Lem. 2.2 of (Cardaliaguet, 2007) proved the existence of $(\alpha_\omega(t), \delta_\omega(t))$ given $(\eta(t), \zeta(t))$.

and the payoff over all game types is $J(t_0, x_0, p; \{\eta_i\}, \zeta) = \mathbb{E}_{i \sim p}[J_i]$. We say the game has a value $V$ if and only if the upper value $V^+(t_0, x_0, p) = \inf_{\{\eta_i\}} \sup_\zeta \mathbb{E}_{\eta_i, \zeta, i}[J_i]$ and the lower value $V^-(t_0, x_0, p) = \sup_\zeta \inf_{\{\eta_i\}} \mathbb{E}_{\eta_i, \zeta, i}[J_i]$ are equal: $V = V^+ = V^-$. $(\{\eta_i\}, \zeta)$ is a Nash equilibrium (NE) if it attains $V$. We introduce the following assumptions under which the game has a value (Cardaliaguet, 2007):

A1. $\mathcal{U} \subseteq \mathbb{R}^{d_u}$ and $\mathcal{V} \subseteq \mathbb{R}^{d_v}$ are compact and finite-dimensional sets.

A2. $f : \mathcal{X} \times \mathcal{U} \times \mathcal{V} \to \mathcal{X}$ is bounded, continuous, and uniformly Lipschitz continuous with respect to $x$.

A3. $g_i : \mathcal{X} \to \mathbb{R}$ and $l_i : \mathcal{U} \times \mathcal{V} \to \mathbb{R}$ are Lipschitz continuous and bounded.

A4. Isaacs' condition holds for the Hamiltonian $H : \mathcal{X} \times \mathbb{R}^{d_x} \to \mathbb{R}$:

$$\begin{aligned} H(x, \xi) &:= \min_{u \in \mathcal{U}} \max_{v \in \mathcal{V}} f(x, u, v)^\top \xi - l_i(u, v) \\ &= \max_{v \in \mathcal{V}} \min_{u \in \mathcal{U}} f(x, u, v)^\top \xi - l_i(u, v). \end{aligned} \tag{2}$$

A5. Both players have full knowledge about $f$, $\{g_i\}_{i=1}^I$, $\{l_i\}_{i=1}^I$, $p_0$, and the Nash equilibrium of the game. Control inputs and states are fully observable and we assume perfect recall.

**Dynamic programming (DP) for P1.** To approximate P1's equilibrium strategy, we introduce a discrete-time value approximation $V_\tau$, which satisfies the following DP (Cardaliaguet, 2009):

$$\begin{aligned} V_\tau(t_0, x_0, p) = \min_{\{\eta_i\}} \mathbb{E}_{u \sim \bar{\eta}}\Big[ &\max_{v \in \mathcal{V}} V_\tau(t_0 + \tau, x'(u, v), p'(u)) \\ &+ \tau \mathbb{E}_{i \sim p'(u)}[l_i(u, v)] \Big], \end{aligned} \tag{3}$$

with a terminal boundary $V_\tau(T, x_0, p) = \sum_i p[i]g_i(x_0)$. Here $x'(u, v)$ solves Eq. 1 starting from $x_0$ for a time span of $\tau$ using constant control inputs $(u, v)$ during $[t_0, t_0 + \tau)$, and $p'(u)$ is the Bayes update of the public belief after P1 takes and P2 observes $u$: $p'(u)[i] = \eta_i(u)p[i]/\bar{\eta}(u)$, where $\bar{\eta}$ is the marginal distribution over $\mathcal{U}$ across types: $\bar{\eta}(u) = \sum_{i \in [I]} \eta_i(u)p_0[i]$. Note that P2's equilibrium cannot be derived from Eq. 3.

**Dual DP for P2.** To compute P2's equilibrium strategy, we need another DP that involves P2's behavioral strategies and P1's best responses. This can be achieved by introducing the convex conjugate $V^*$ of $V$:

$$V^*(t_0, x_0, \hat{p}) := \max_p p^T \hat{p} - V(t_0, x_0, p)$$

$$= \max_p p^T \hat{p} - \sup_{\zeta \in \mathcal{Z}_r(t_0)} \inf_{\{\eta_i\} \in \{\mathcal{H}_r(t_0)\}^I} \mathbb{E}_{\eta_i, \zeta, i}\Big[J_i(t_0, x_0; \eta_i, \zeta)\Big]$$

$$= \max_p \inf_{\zeta \in \mathcal{Z}_r(t_0)} \sup_{\{\eta_i\} \in \{\mathcal{H}_r(t_0)\}^I} p^T \hat{p} - \mathbb{E}_{\eta_i, \zeta, i}\Big[J_i(t_0, x_0; \eta_i, \zeta)\Big]$$

$$= \inf_{\zeta \in \mathcal{Z}_r(t_0)} \sup_{\eta \in \mathcal{H}(t_0)} \max_{i \in \{1, ..., I\}} \left\{\hat{p}_i - \mathbb{E}_\zeta\Big[J_i(t_0, x_0; \eta_i, \zeta)\Big]\right\}. \tag{4}$$

The last step of Eq. 4 uses the linearity of the payoff with respect to $p$ and again the fact that best responses are always pure (thus $\eta$ belongs to the pure strategy set $\mathcal{H}(t_0)$ rather than the random strategy set $\mathcal{H}_r(t_0)$). Eq. 4 describes a dual game with complete information, where the strategy space of P1 becomes $\mathcal{H}(t_0) \times [I]$, i.e., the game type is now chosen by P1 rather than the nature. It is proved that P2's equilibrium in the dual game is also an equilibrium for the primal game if $\hat{p} \in \partial_p V(t_0, x_0, p)$. We explain in App. D that such $\hat{p}$ represents the type-dependent gains of P1 should he play the best responses to P2's equilibrium strategy. Therefore $\hat{p}_i - \mathbb{E}_\zeta[g_i + \int l_i]$ measures P2's risk and his equilibrium strategy is to minimizes the worst-case risk across all game types. The DP of P2 in this dual game is (Cardaliaguet, 2009):

$$V_\tau^*(t_0, x_0, \hat{p}) =$$

$$\min_{\zeta, \hat{p}'(v)} \mathbb{E}_{v\sim\zeta}\left[ \max_{u\in\mathcal{U}} V_\tau^*(t_0 + \tau, x'(u,v), \hat{p}'(v) - \tau l(u,v)) \right], \tag{5}$$

with a terminal boundary $V^*(T, x_0, \hat{p}) = \max_{i\in[I]}\{\hat{p}[i] - g_i(x_0)\}$. Here $\hat{p}'(v) : \mathcal{V} \to \mathbb{R}^I$ is constrained by $\mathbb{E}_{v\sim\zeta}[\hat{p}'(v)] = \hat{p}$, and $l(u,v)[i] = l_i(u,v)$.

Let P1's strategy set from Eq. 3 be $\{\eta_{i,\tau}\}$ and P2's from Eq. 5 be $\zeta_\tau$. Thm. 3.1 proves that $(\{\eta_{i,\tau}\}, \zeta_\tau)$ approaches the equilibrium of $V$ when $\tau$ is sufficiently small (App. A completes the proof sketch in Cardaliaguet (2009)):

**Theorem 3.1.** *(Thm.4.1 of Cardaliaguet (2009)) If A1-5 hold, then there exists some $M_1, M_2 > 0$, such that $V(t_0, x_0, p) \le \max_{\zeta\in\mathcal{Z}(t_0)} J(t_0, x_0, p; \{\eta_{i,\tau}\}, \zeta) \le V(t_0, x_0, p) + M_1(T - t_0)\tau$ for any $(t_0, x_0, p) \in [0,T] \times \mathcal{X} \times \Delta(I)$, and $V^*(t_0, x_0, \hat{p}) \le \max_{\{\eta_i\}\in\{\mathcal{H}^i\}^I} J^*(t_0, x_0, \hat{p}; \{\eta_i\}, \zeta_\tau) \le V^*(t_0, x_0, \hat{p}) + M_2(T - t_0)\tau$ for any $(t_0, x_0, \hat{p}) \in [0,T] \times \mathcal{X} \times \mathbb{R}^I$.*

**Remarks.** Notice that the DPs consider conservative approximations of the original game. E.g., the primal DP considers P2 play the best responses to the actions *to be played* by P1, thus $V(t_0, x_0, p) \le \max_{\zeta\in\mathcal{Z}(t_0)} J(t_0, x_0, p; \{\eta_{i,\tau}\}, \zeta)$. Nonetheless, by using continuity and boundedness assumptions (A1-3) and Isaacs' condition (A4), Thm. 3.1 shows that the advantages taken by best responses in the DPs are limited. Importantly, approximating the original game through the DPs enables the "splitting" reformulation that critically addresses the scalability issue with respect to continuous action spaces, which we discuss in Sec. 4.

## 4. A Splitting Reformulation of the DPs

With Thm. 3.1, we can approximate P1's strategy by solving Eq. 3, and P2's by solving both Eq. 3 and Eq. 5 because his strategy depends on $\partial_p V(t_0, x_0, p)$. These minimax problems need to be solved at sufficiently many collocation points $((t, x, p)$ or $(t, x, \hat{p}))$ and with a sufficiently refined

time discretization. In the context of IIEFGs, both DPs can be considered as sequential games where the leader plays a mixed strategy and the follower a best response. Existing algorithms, e.g., CFR+, CFR-BR, and MMD, are not scalable at solving the DPs when the games have continuous action spaces and many time steps. To this end, our key insight is the following theorem, which states that P1's strategy that solves the primal DP is $I$-atomic and P2's is $(I + 1)$-atomic (proof in App. B):

**Theorem 4.1.** *The RHS of Eq. 3 can be reformulated as*

$$\min_{\{u^k\}, \{\alpha_{ki}\}} \max_{\{v^k\}} \sum_{k=1}^{I} \lambda^k \Big( V(t + \tau, x^k, p^k)$$
$$+ \tau\mathbb{E}_{i\sim p^k}[l_i(u^k, v^k)] \Big)$$

$$s.t. \quad u^k \in \mathcal{U}, \quad x^k = ODE(x, \tau, u^k, v^k; f), \quad v^k \in \mathcal{V}, \quad (\text{P}_1)$$

$$\alpha_{ki} \in [0,1], \sum_{k=1}^{I} \alpha_{ki} = 1, \lambda^k = \sum_{i=1}^{I} \alpha_{ki} p[i],$$

$$p^k[i] = \frac{\alpha_{ki} p[i]}{\lambda^k}, \quad \forall i, k \in [I].$$

*And the RHS of Eq. 5 can be reformulated as*

$$\min_{\{v^k\}, \{\lambda^k\}, \{\hat{p}^k\}} \max_{\{u^k\}} \sum_{k=1}^{I+1} \lambda^k \Big( V^*(t + \tau, x^k, \hat{p}^k - \tau l(u^k, v^k)) \Big)$$

$$s.t. \quad u^k \in \mathcal{U}, \quad v^k \in \mathcal{V}, \quad x^k = ODE(x, \tau, u^k, v^k; f),$$

$$\lambda^k \in [0,1], \sum_{k=1}^{I+1} \lambda^k \hat{p}^k = \hat{p}, \sum_{k=1}^{I+1} \lambda^k = 1, \ k \in [I+1].$$
$$(\text{P}_2)$$

**Sketch of the proof.** By change of variable and introducing a pushforward measure, we can show that the RHS of the primal (resp. dual) DP essentially seeks a mixed strategy that convexifies the value (resp. dual value) at the next time step over $\Delta(I)$ (resp. $\mathbb{R}^I$). Since convexification requires at most $I$ vertices in $\Delta(I)$ (resp. $I + 1$ vertices in $\mathbb{R}^I$), the resultant strategy is at most $I$-atomic (resp. $(I+1)$-atomic).

**A visual example.** Fig. 2 provides an intuitive explanation of the causality between value convexification and the equilibrium strategy, where the public belief $p \in \Delta(2)$: Let the solid red line be $U_\tau(t_0, x_0, p) := \min_u \max_v V(t_0 + \tau, x'(u,v), p) + \tau\mathbb{E}_i[l_i(u,v)]$. We call $U_\tau(t_0, x_0, p)$ the value of a *non-revealing* version of the game because $p$ does not change over the course of this game when P1 plays pure. One notices that if $U_\tau$ is not convex in $p$, it is always possible for P1 to achieve a lower value by convexifying $U_\tau$ through the use of a mixed strategy, leading to $\text{P}_1$. In this particular case, P1 identifies $[\lambda^a, \lambda^b]^T \in \Delta(2)$ and $\{p^a, p^b\}$ such that $\lambda^a p^a + \lambda^b p^b = p$. Picking one action $u^k \in \arg\min_u \max_v V(t_0 + \tau, x'(u,v), p^k) + \tau\mathbb{E}_i[l_i(u,v)]$ for each $k \in \{a, b\}$[2], P1 of type $i$ will then play action $u^k$ with probability $\alpha_{ki} = p^k[i]\lambda^k/p[i]$. By announcing this

---

[2]Isaacs' condition guarantees that $\min_u \max_v V(t_0 + \tau, x'(u,v), p^k) + \tau\mathbb{E}_i[l_i(u,v)]$ has a solution.

strategy, the public belief shifts to $p^k$ via the Bayes' rule if P1 takes action $u^k$, and as a result, P1 receives a value $V(t_0, x_0, p) = \lambda^a U(t_0, x_0, p^a) + \lambda^b U(t_0, x_0, p^b)$, which is the convexification of $U(t_0, x_0, p)$ over $p \in \Delta(2)$. The same splitting happens for P2 in the dual game: instead of the public belief $p$, P2's strategy splits the dual variable $\hat{p}$ to $\hat{p}^k$ by playing action $v^k$ with probability $\lambda^k$. We note that this convexification nature of the equilibrium strategies has been discovered as the "Cav u" theorem as early as for 2p0s repeated games with one-sided information (Aumann et al., 1995; De Meyer, 1996). Our new contribution is in explaining its connection with IIEFGs (see below) and in developing a scalable algorithm for value and strategy approximation that takes advantage of this property along with multigrid (see Sec. 5).

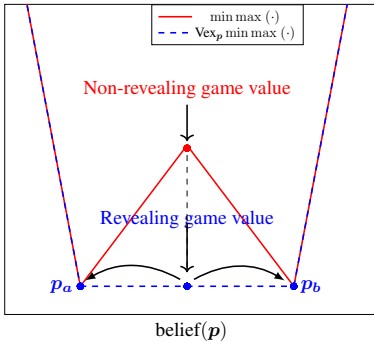

*Figure 2.* Revealing and non-revealing game values, and the mechanism of splitting.

**Comparison with CFR-BR.** For conciseness, we introduce CFR-BR as a representative IIEFG algorithm to compare with $P_1$ and $P_2$, since CFR-BR also decouples the solving of P1 and P2's strategies by letting one player always play the best response to the opponents' behavioral strategy. In the context of this paper, CFR-BR solves

$$V_\tau(t_0, x_0, p) =$$
$$\min_{\{\eta_i\}} \max_{v \in \mathcal{V}} \mathbb{E}_u \left[ V_\tau(t_0 + \tau, x'(u, v), p'(u)) + \tau \mathbb{E}_{i \sim p'(u)}[l_i] \right] \quad (6)$$
$$= \max_{\zeta} \min_{u \in \mathcal{U}} \mathbb{E}_v \left[ V_\tau(t_0 + \tau, x'(u, v), p) + \tau \mathbb{E}_{i \sim p}[l_i] \right].$$

We first note that the CFR-BR formulation does not enjoy atomic mixed strategies as in the DPs. This is because the best responses of P2 are upon the mixed strategies of P1 rather his actual actions. Therefore the non-revealing value $U_\tau(t_0, x_0, p)$ is implicitly a function of P1's mixed strategies rather than of a single action. As a result, the RHSs of Eq. 6 cannot be rewritten as convexification over public belief. This causes CFR-BR to suffer from slow convergence when fine discretization of the continuous action spaces is necessary. On the other hand, using a leader-follower reformulation of the game, $P_1$ and $P_2$ reveals the atomic nature of the equilibrium strategies via Thm. 3.1 and Thm. 4.1. We note that the leader-follower formulation in Thm. 4.1 is applicable to the following game settings with one-sided payoff information:

1. differential games where A1-3 make up for the incorrect leader-follower setting (this paper, and see App. E.1 for an analytical example where $P_1$ and $P_2$ correctly solve a differential game),
2. turn-based extensive-form games where the assignment of leader and follower is naturally correct (see App. E.2 for the derivation of the Nash equilibrium using $P_1$ and $P_2$ for a turn-based game), and
3. infinitely repeated normal-form games where splitting only happens in the first step of the game for which the payoff does not affect the value (see De Meyer (1996)).

---

**Algorithm 1** CAMS for P1

**Input:** time discretization $\tau$, terminal value $V(T, \cdot, \cdot)$, sample size $N$, minimax solver $\mathbb{O}$
**Initialize:** value network $\{\hat{V}_t\}_{t=0}^{T-\tau}$, training dataset $\mathcal{S} \leftarrow \emptyset$
$\mathcal{S} \leftarrow$ sample $N$ states $(x, p) \in \mathcal{X} \times \Delta(I)$
  **for** $t$ *in* $\{T - \tau, ..., 0\}$ **do**
    **for** $(x, p)$ *in* $\mathcal{S}$ **do**
      $\vartheta \leftarrow \mathbb{O}(t, x, p)$ ;    /* Solution to P1 */
      append $\{(t, x, p), \vartheta\}$ to $\mathcal{S}$
    Fit $\hat{V}_t$ to $\mathcal{S}$

---

**The proposed algorithm.** We discretize the time span $[0, T]$ as $\{k\tau\}_{k=0}^K$ where $\tau = T/K$, and denote by $\mathcal{S} = \{(x, p)_i\}_{i \in [|\mathcal{S}|]}$ and $\mathcal{S}^* = \{(x, \hat{p})_i\}_{i \in [|\mathcal{S}^*|]}$ the primal and dual sample set, respectively. The backward induction solves $P_1$ (resp. $P_2$) starting from $t = (K - 1)\tau$ at all collocation points in $\mathcal{S}$ (resp. $\mathcal{S}^*$). The resultant nonconvex-nonconcave minimax problems have size $(\mathcal{O}(I(I + d_u)), \mathcal{O}(Id_v))$ (resp. $(\mathcal{O}(I(I + d_v)), \mathcal{O}(Id_u))$). Importantly, the computational complexity of these problems are no longer related to the size of the action spaces. To generalize value (and optionally policy) prediction across the continuous joint space of state and belief, primal and dual value networks are trained on the minimax solutions. The value networks are used to formulate the next round of minimax at $t - \tau$. The backward induction continuous until $t = 0$. Alg. 1, dubbed CAMS (Continuous Action Mixed Strategy solver), summarizes the proposed algorithm for P1.

**The remaining computational challenges.** Our discussion so far addresses the scalability issue due to large or continuous action spaces. In particular, when the number of possible game types is small, i.e., $I^2 \ll |\mathcal{U}| + |\mathcal{V}|$, solving $P_1$ and $P_2$ becomes more efficient than using IIEFG solvers. The computational challenge, however, still remains for two reasons: (1) Thm. 3.1 suggests a fine enough time discretization for the strategies derived from $P_1$ and $P_2$ to be good approximations of the equilibrium. (2) Through the baseline algorithm, suppressing the $L_\infty$ value prediction error at $t = 0$ requires a computational complexity exponential to the number of time steps $K$. Specifically, let $\hat{V}_0(x, p) : \mathcal{X} \times \Delta(I) \to \mathbb{R}$ be the trained value networks at $t = 0$, we have the following result (see proof in App. G):

**Theorem 4.2.** *Given the number of time steps $K$, a minimax approximation error $\epsilon > 0$, a prediction error threshold $\delta > 0$, there exists some constant $C \geq 1$, such that with a computational complexity of at least $\mathcal{O}(K^3 C^{2K} I^2 \epsilon^{-4} \delta^{-2})$, Alg. 1 achieves*

$$\max_{(x,p)\in\mathcal{X}\times\Delta(I)} |\hat{V}_0(x,p) - V(0,x,p)| \leq \delta. \tag{7}$$

A similar result applies to the dual game. Zanette et al. (2019) discussed a linear value approximator that achieves $C = 1$. However, their method requires solving a linear program (LP) for every inference $\hat{V}_t(x,p)$ if $(x,p)$ does not belong to the training set $\mathcal{S}$. In our context, incorporating their method would require auto-differentiating through the LP solver during each descent and ascent steps in solving the minimax problems, which turned out to be expensive in PyTorch and JAX. While effective suppression of $C$ for neural nets remains to be investigated, this paper introduces a multigrid approach to reduce the cost for games with a large $K$, as we discuss in Sec. 5.

## 5. A Multigrid Approach

We introduce a multigrid approach that accelerates value approximation through backward inductions on multiple time grids. Since strategies at time $t$ are implicitly nonlinear functions of the value at $t + \tau$, the primal and dual HJI PDEs underlying $P_1$ and $P_2$ are nonlinear. Therefore, our method will extend the Full Approximation Scheme (FAS) commonly used for solving nonlinear PDEs, where PDEs are solved on all grids and coarse-grid corrections are then used to improve fine-grid solutions (Trottenberg et al., 2000; Henson et al., 2003). In theory, FAS reduces the required number of "fine sweeps" by shifting global error correction onto the cheaper coarse pass. More concretely, a two-grid FAS has four steps (see illustration in Fig. 1(b)): (1) Restrict the fine-grid approximation and its residual; (2) solve the coarse-grid problem using the fine-grid residual; (3) compute the coarse-grid correction; (4) prolong the coarse-grid correction to fine-grid and add the correction to fine-grid approximation.

For conciseness, we will focus on the primal problem to introduce the FAS extension. Let $\hat{V}_t^l$ be the value network for time $t$ on grid size (time interval) $l$. Let the restriction operators be $\mathcal{R}^l$ from a finer grid with grid size $l$ to a coarser one with size $2l$: $\mathcal{R}^l(\hat{V}_t^l) = (\hat{V}_t^l + \hat{V}_{t+l}^l)/2$ is the value restriction from $l$ to $2l$. This restriction operator takes into account the backward induction nature of value functions. Similarly, we define the prolongation operators $\mathcal{P}^{2l}$ as:

$$\mathcal{P}^{2l}(\hat{V}_t^{2l}) = \begin{cases} \hat{V}_t^{2l}, & \text{if } t \in \mathcal{T}^{2l} \\ \hat{V}_{t+l}^{2l}, & \text{otherwise} \end{cases}, \tag{8}$$

where $\mathcal{T}^{2l} := \{n \cdot 2l : n \in \mathbb{N}_0, n < T/2l\}$. Let $\mathbb{O}^l(t,x,p;\hat{V})$ solves $P_1$ at $(t,x,p)$ using $\tau = l$ and $\hat{V}$ as the value at $t + \tau$, and outputs an approximation for $V(t,x,p)$. The dataset $\{(t,x^{(j)},p^{(j)},\mathbb{O}^l(t,x^{(j)},p^{(j)};\hat{V}_{t+l}^l))\}$ is used

to train $\hat{V}_t^l(\cdot,\cdot)$. Let $r_t^l(x,p) = \hat{V}_t^l(x,p) - \mathbb{O}^l(t,x,p;\hat{V}_{t+l}^l)$ be the residual. On each grid, our goal is to find $\hat{V}_t^l$ such that $r_t^l(x,p) \approx 0$ for all $(t,x,p) \in \mathcal{T}^l \times \mathcal{X} \times \Delta(I)$. This is achieved by restricting the fine grid approximations and residuals to the coarse grid and solving to determine the corrections. Let $e_t^l(x,p)$ be the correction in grid $l$ at $(t,x,p)$. Then, the coarse-grid problem is:

$$\underbrace{\mathcal{R}^l r_t^l}_{\text{residual}} = \overbrace{\mathbb{O}^{2l}(t,x,p;\mathcal{R}^l \hat{V}_{t+2l}^l + e_{t+2l}^{2l}) - \left(\mathcal{R}^l \hat{V}_t^l + e_t^{2l}(x,p)\right)}^{\text{coarse-grid eq. w/ corrections}}$$
$$- \underbrace{\left(\mathbb{O}^{2l}(\mathcal{R}^l \hat{V}_{t+2l}^l) - \mathcal{R}^l \hat{V}_t^l\right)}_{\text{coarse-grid eq. w/o corrections}}, \tag{9}$$

which is solved backward from $T - 2l$, as the terminal value is known, resulting in $e_T^{2l} = 0$. Knowing $e_{t+2l}^{2l}(\cdot,\cdot)$, the FAS coarse-grid correction at $(t,x,p)$ is:

$$e_t^{2l}(x,p) = \mathbb{O}^{2l}(t,x,p;\mathcal{R}^l \hat{V}_{t+2l}^l + e_{t+2l}^{2l})$$
$$- \mathbb{O}^{2l}(\mathcal{R}^l \hat{V}_{t+2l}^l) - \mathcal{R}^l r_t^l. \tag{10}$$

This correction ensures consistency: If $\hat{V}_t^l = V(t,\cdot,\cdot)$ for all $t \in \mathcal{T}^l$, $e_t^{2l}(\cdot,\cdot) = 0$ for all $t \in \mathcal{T}^{2l}$. The coarse grid corrections are prolonged to the fine grid to update the fine-grid value approximation. Alg. 2 summarizes a 2-level multigrid algorithm, and Alg. 3 for an $n$-level version (see App. J). Note that from Eq. 10, computing the coarse correction in our case requires two separate minimax calls with similar loss formulations. We further accelerate the multigrid solver by warm-starting these minimax problems using the recorded minimax solution derived from the fine grid (during the residual computation).

## 6. Empirical Validation

We introduce Hexner's game (Hexner, 1979) that has an analytical Nash equilibrium. We use variants of this game to compare CAMS with existing baselines (MMD, CFR+, JPSPG, and DeepCFR) on solution quality and computational cost. We also demonstrate the scalability of CAMS using a high-dimensional version of the game in App. K.

### 6.1. Hexner's game

In Hexner's game, the dynamics is decomposed as $\dot{x}_j = A_j x_j + B_j u_j$ for $j = [2]$, where $x_j \in \mathcal{X}_j$, $u_j \in \mathcal{U}_j$, and $A_j$ and $B_j$ are known matrices. The target state of P1 is $z\theta$ where $\theta$ is drawn with distribution $p_0$ from $\Theta$, $|\Theta| = I$, and $z \in \mathbb{R}^{d_x}$ is fixed and common knowledge. Denote by $\eta_i(t)$ and $\zeta(t)$ the random actions at time $t$ induced by strategy pair $(\eta_i, \zeta)$. The expected payoff to P1 is:

$$J(\{\eta_i\}, \zeta) = \mathbb{E}_{i \sim p_0}\left[\int_0^T (\eta_i(t)^\top R_1 \eta_i(t) - \zeta(t)^\top R_2 \zeta(t))dt\right.$$
$$+ [x_1(T) - z\theta_i]^\top K_1(T)[x_1(T) - z\theta_i]$$
$$\left. - [x_2(T) - z\theta_i]^\top K_2(T)[x_2(T) - z\theta_i]\right], \tag{11}$$

where $R_1, R_2 \succ 0$ are control-penalty matrices and $K_1, K_2 \succeq 0$ are state-penalty matrices. Essentially, the goal of P1 is to get closer to the target $z\theta$ than P2. To take full information advantage, P1 needs to decide when to home-in to and thus reveal the target. See Fig. 1(c) for an illustration. As explained in Hexner (1979) and Ghimire et al. (2024), this game has an analytical solution: There exists a problem-dependent critical time $t_r := t_r(T, \{A_j\}, \{B_j\}, \{R_j\}, \{K_j\})$, if $t_r \in (0, T)$, P1 homes towards the mean target $\mathbb{E}[\theta]$ as if he does not know the actual target until $t_r$. If $t_r \leq 0$, P1 homes towards the actual target at $t = 0$. P2's strategy is to follow P1.

## 6.2. Comparisons on 1- and 4-stage Hexner's games

**Settings.** We first use a normal-form Hexner's game with $\tau = T$ and a fixed initial state $x_0 \in \mathcal{X}$ to demonstrate that baseline algorithms suffer from increasing costs along the size of the discrete action space while CAMS does not. The baselines we consider include CFR+ (Tammelin, 2014), MMD (Sokota et al., 2022), Joint-Perturbation Simultaneous Pseudo-Gradient (JPSPG) (Martin & Sandholm, 2024), and a modified CFR-BR (Johanson et al., 2012) (dubbed CFR-BR-Primal), where we only compute P2's best response to P1's current strategy and only focus on converging P1's strategy, which matches with CAMS for solving $P_1$. Among these, only JPSPG can handle continuous action spaces. All baselines (except JPSPG) are implemented in OpenSpiel (Lanctot et al., 2019). The normal-form primal game has a trivial ground-truth strategy where P1 goes directly to his target. For visualization, we use $d_x = 4$ (position and velocity in 2D). For baselines (except JPSPG), we use discrete action sets defined by 4 lattice sizes so that $U = |\mathcal{U}_j| \in \{16, 36, 64, 144\}$. All algorithms terminate when a threshold of NashConv[3] is met. For conciseness, we only consider solving P1's strategy and thus use P1's $\delta$ in NashConv. We set the threshold to $10^{-3}$ for baselines and $10^{-5}$ for CAMS. We will show that even with a more stringent threshold, CAMS still converges significant faster than the baselines. We then use DeepCFR and JPSPG as baselines for a Hexner's game with 4 time steps, where $T = 1$ and $\tau = 0.25$. DeepCFRs were run for 1000 CFR iterations (resp. 100) with 10 (resp. 5) traversals for $U = 9$ (resp. 16). More details on experiment settings can be found in App. H.3. Similarly, JPSPG was run for $2 \cdot 10^8$ iterations, where each iteration consisted of solving a game with a random initial state and type, and performing a strategy update. More details in App. H.4.

**Comparison metrics.** For the normal-form game, we compare both computational cost and the expected action error $\varepsilon$ from the ground-truth action of P1: $\varepsilon(x_0) := \mathbb{E}_{i \sim p_0} \left[ \sum_{k=1}^{|\mathcal{U}|} \alpha_{ki} \|u_k - u_i^*(x_0)\|_2 \right]$, where $u_i^*(x_0)$ is the ground truth for type $i$ at $x_0$. For the 4-stage game, we compare the expected action errors at each time step:

[3]See Lanctot et al. (2017) for a definition of NashConv.

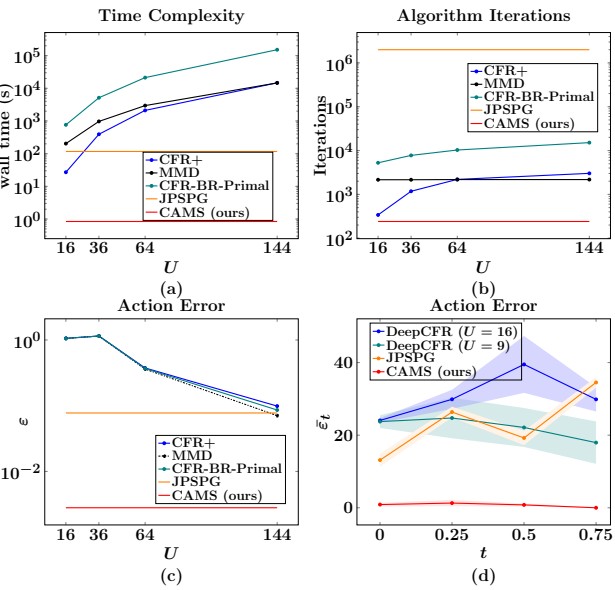

*Figure 3.* (a-c) Comparisons b/w CAMS (ours), JPSPG, CFR+, MMD, CFR-BR-Primal on 1-step Hexner's game. (d) Comparison b/w CAMS, JPSPG, and DeepCFR on 4-stage Hexner's w/ similar compute.

$\bar{\varepsilon}_t := \mathbb{E}_{x_t \sim \pi}[\varepsilon(x_t)]$, where $\pi$ is the strategy learned by DeepCFR, JPSPG, or CAMS. For each strategy, we estimate $\{\bar{\varepsilon}_t\}_{t=1}^4$ by generating 100 trajectories with initial states uniformly sampled from $\mathcal{X}$. The wall-time costs for game solving are 17 hours using CAMS (baseline), 24 hours for JPSPG, 29 hours ($U = 9$) and 34 hours ($U = 16$) using DeepCFR, all on an A100 GPU.

**Results.** Fig. 3 summarizes the comparisons. For the normal-form game, all baselines (except JPSPG) have complexity and wall-time costs increasing with $U$, while CAMS is invariant to $U$. With the similar or less compute, CAMS achieves significantly better strategies than DeepCFR and JPSPG in the 4-stage game. Sample trajectories for the 4-stage game are shown in App. H.

## 6.3. Scalability of CAMS

**10-stage game.** Here we solve Hexner's games with $T = 1$ and $\tau = 0.1$, and consider both state-constrained and unconstrained cases. These games have a game-tree complexity of $10^{80}$ if we use an action discretization of $U = 10k$ (100 discrete values along each of the two action dimensions). In the state-constrained version of the game, P1 receives $+\infty$ if he collides with P2. Collision occurs when the Euclidean distance between the players is less than 0.05. As a result, the Nash equilibrium of this game variant is no longer analytical. Following (Ghimire et al., 2024), we approximate a time-dependent safe zone $\Omega_t \subseteq \mathcal{X}$ for P1 so that for any initial state outside of $\Omega_t$, P1 surrenders because P2 can always find a strategy to collide. Within $\Omega_t$, the Nash equilibrium can be derived from CAMS where for each minimax problem, P1's admissible actions are restricted by $\Omega_t$. In (Ghimire et al., 2024), the resultant constrained

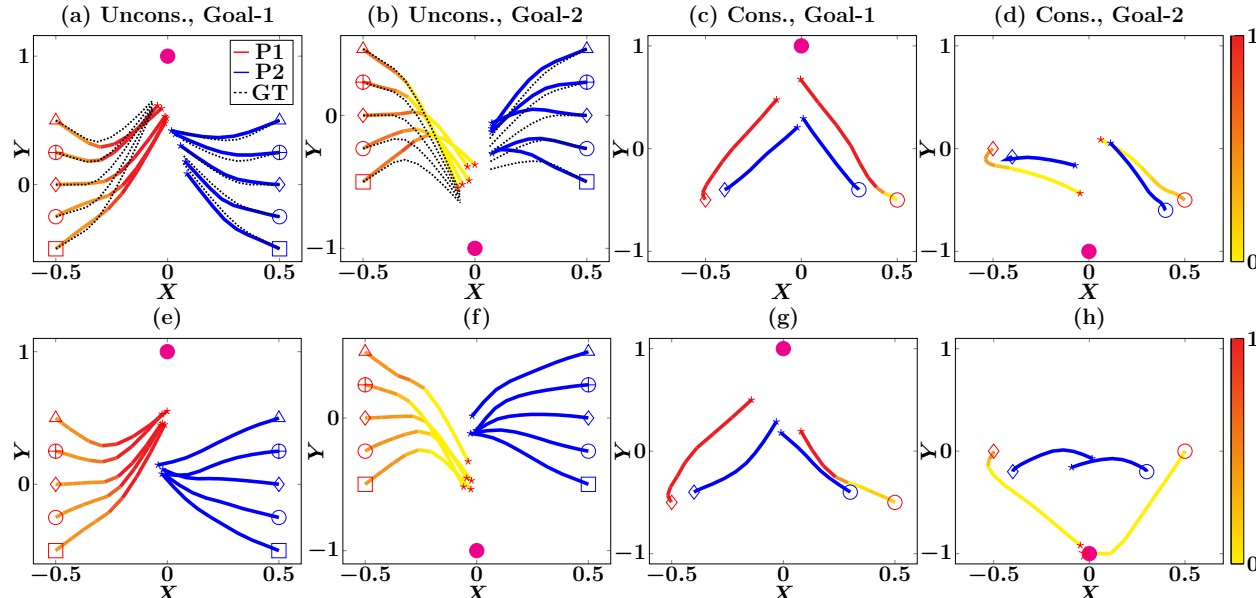

*Figure 4.* Sample trajectories for the primal game (a-d) where P1 plays Nash and P2 plays best response, and primal-dual game (e-h) where both players play Nash. Cols 1 & 2 are unconstrained, cols 3 & 4 are w/ collision constraint. Dotted lines are ground-truth Nash. Color shades indicate evolution of public belief (1 means Goal-1). Initial position pairs are marked with same markers.

minimax problems are solved as follows: First, at each $t$, non-revealing games (without splitting) are approximately solved across $\mathcal{X} \times \Delta(I)$ using an enumeration over $U = 100$. This requires finding the minimax point from a $100 \times 100$ matrix for each $(x, p)$. Then with the resultant values for the non-revealing games, the convex hull of the value over the public belief is approximated for each sampled $x$, before fitting a neural network $\hat{V}_t$ to these approximated convex hulls. Due to the use of enumeration, this method has exponential space and computational complexities with respect to the dimensionalities of the action spaces. In this paper, we solve $P_1$ and $P_2$ which directly approximates the convexified values. In addition, since DS-GDA is gradient-based, the resultant space and computational complexities are only linear to the dimensionality of the action spaces.

**Results:** Results are summarized in Fig. 4. For the unconstrained game where analytical strategies are known, we compare the approximated and the ground-truth strategies starting from various initial states. While approximation errors exist, CAMS successfully learns the target-concealing behavior of P1 as P1 always moves towards $\mathbb{E}_{p_0}[z\theta]$ before revealing his target. Averaging over 50 trajectories derived from CAMS, P1 conceals the target until $t_r = 0.60s \pm 0.06s$ (compared to the ground-truth $t_r = 0.5s$). CAMS also approximates P2's robust strategy well, as P2 only starts to home towards a target after P1 reveals. We note that the complexity of the dual game is higher than that of the primal game because its value is one dimension higher and $P_2$ is larger than $P_1$. This resulted in higher error in approximating P2's strategies.

### 6.4. Accelerating value approximation with multigrid

Here we demonstrate the efficacy of multigrid methods (see Alg. 2 and Alg. 3 in App. J) in accelerating value function approximation. We report the runtime[4] of all algorithms (Algs. 1, 2, 3) on 4-, 10-, and 16-stage games in Tab. 2. We run the 2-level multigrid (Alg. 2) on the 4- and 10-stage games, and 4-level multigrid (Alg. 3) on the 16-stage game. We also report the resulting trajectories in App. J.

*Table 2.* Runtime Comparison: CAMS w/ and w/o Multigrid

| # time steps | w/o multigrid | w/ multigrid ↓ |
|---|---|---|
| 4 | 9.3 hrs | 2.32 hrs |
| 10 | 27.6 hrs | 10.9 hrs |
| 16 | 46.21 hrs | 17.83 hrs |

## 7. Conclusion

Unlike IIEFGs where mixed strategies have to be approximated over the entire action space across the game tree, we showed that differential games with one-sided payoff information enjoy a much simpler strategy structure when the Isaacs' condition holds: The strategy of the informed (resp. uninformed) player has at most $I$ (resp. $I + 1$) pure action branches at each infostate. We demonstrated the clear advantage of using this structural property in solving games with continuous action spaces, against SOTA IIEFG solvers, in terms of computational cost and solution quality. We also showed that multigrid further accelerates value and strategy approximation. To the authors' best knowledge, this is the first method to provide tractable solution for incomplete-information games with continuous action spaces without problem-specific abstraction and discretization.

---

[4]Experiments done on one H100 GPU

## Impact Statement

This work is concerned with bridging the gap between computational game theory and differential game theory. With its possible applications to robotics and AI, there is a need for studies on mitigating risks arising from deceptive strategies by robots and machines against human beings.

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

## A. Proof of Theorem 3.1

**Theorem 3.1** For any $(t_0, x_0, p) \in [0, T] \times \mathcal{X} \times \Delta(I)$, if A1-5 hold, then there exist $M_1, M_2 > 0$, such that $V(t_0, x_0, p) \leq \max_{\zeta \in \mathcal{Z}(t_0)} J(t_0, x_0, p; \{\eta_{i,\tau}\}, \zeta) \leq V(t_0, x_0, p) + M_1(T - t_0)\tau$. Similarly, for any $(t_0, x_0, \hat{p}) \in [0, T] \times \mathcal{X} \times \mathbb{R}^I$, $V^*(t_0, x_0, \hat{p}) \leq \max_{\{\eta_i\} \in \{\mathcal{H}^i(t_0)\}^I} J^*(t_0, x_0, \hat{p}; \{\eta_i\}, \zeta_\tau) \leq V^*(t_0, x_0, \hat{p}) + M_2(T - t_0)\tau$.

*Proof.* Our proof completes the sketch for Theorem 4.1 in Cardaliaguet (2009). For the primal game, by definition we have $V(t_0, x_0, p) \leq \max_{\zeta \in \mathcal{Z}(t_0)} J(t_0, x_0, p; \{\eta_{i,\tau}\}, \zeta)$. So we just need to prove

$$\max_{\zeta \in \mathcal{Z}(t_0)} J(t_0, x_0, p; \{\eta_{i,\tau}\}, \zeta) \leq V(t_0, x_0, p) + M_1(T - t_0)\tau. \tag{12}$$

For some $(t_0, x_0, p)$, let $t_1 = t_0 + \tau$, and $v^\dagger$ be the ground truth equilibrium action at $(t_0, x_0, p)$. Given some $u \in \mathcal{U}$, let $x^\dagger(u) := X_{t_1}^{t_0, x_0, u, v^\dagger}$. Denote by $v_0$ the solution to $\max_{v \in \mathcal{V}} V(t_1, x'(u, v), p) + L(u, v, p)$ and $x_1 = X_{t_1}^{t_0, x_0, u, v_0}$. And let $L(u, v, p) := \mathbb{E}_{i \sim p}[\int_{t_0}^{t_1} l_i(u, v) ds]$ be the expected running cost in $[t_0, t_1]$. We first show that with some $L_1, L_2 > 0$

$$|V(t_1, x_1, p) + L(u, v_0, p) - V(t_1, x^\dagger(u), p) - L(u, v^\dagger, p)| \leq L_1 L_2 \tau^2. \tag{13}$$

To do this, we note that for a small enough $\tau$,

$$V(t_1, x^\dagger(u), p) = V(t_1, x_1, p) + \nabla_x V|_{x_1}(x^\dagger(u) - x_1) + (x^\dagger(u) - x_1)^\top \nabla_x^2 V|_{x_1}(x^\dagger(u) - x_1)$$
$$= V(t_1, x_1, p) + \nabla_x V|_{x_1} \nabla_v x_1|_{v_0}(v^\dagger - v_0) + (x^\dagger(u) - x_1)^\top \nabla_x^2 V|_{x_1}(x^\dagger(u) - x_1) \tag{14}$$

and

$$L(u, v^\dagger, p) = L(u, v_0, p) + \nabla_v L|_{v_0}(v^\dagger - v_0). \tag{15}$$

From the definition of $v_0$, we have

$$\nabla_v(V + L)|_{v_0} = \nabla_x V|_{x_1} \nabla_v x_1|_{v_0} + \nabla_v L|_{v_0} = 0. \tag{16}$$

Together with the assumptions that $V$ is $L_1$-smooth and the dynamics $f$ is $L_2$-Lipschitz continuous (A2-3), we get Eq. 13. Equivalently, we have

$$V(t_1, x_1, p) + L(u, v_0, p) \leq V(t_1, x^\dagger(u), p) + L(u, v^\dagger, p) + L_1 L_2 \tau^2. \tag{17}$$

Since Eq. 17 holds for any $u \in \mathcal{U}$, we have

$$V(t_1, x_1, p) + L(u, v_0, p) \leq V(t_0, x_0, p) + L_1 L_2 \tau^2. \tag{18}$$

Let $x^u = X_{t_1}^{t_0, x_0, \eta, \zeta(u)}$, $\bar{\eta}_\tau(u) = \sum_{i \in [I]} \eta_{i,\tau}(u) p[i]$ be the marginal of taking action $u$, and $p^u[i] = \eta_{i,\tau}(u) p[i] / \bar{\eta}_\tau(u)$ be the updated belief after observing action $u$.

Now we prove Eq. 12 by backward induction. At $t = T$, Eq. 12 holds due to the terminal boundary. Let us assume that it holds true for some $t_1 = t_0 + \tau \in (0, T]$. Let $(x_0, p)$ be fixed. For any $\zeta \in \mathcal{Z}(t_0)$, we have

$$J(t_0, x_0, p; \{\eta_{i,\tau}\}, \zeta) = \sum_{i \in [I]} p[i] \mathbb{E}_{\{\eta_{i,\tau}\}}[J_i(t_0, x_0; \eta_{i,\tau}, \zeta)]$$

$$= \sum_{i \in [I]} p[i] \int_{\mathcal{U}} \eta_{i,\tau}(u) \Big( \mathbb{E}_{\{\eta_{i,\tau}^u\}}[J_i(t_1, x^u; \eta_{i,\tau}^u, \zeta^u)] + \int_{t=t_0}^{t_1} l_i(u, \zeta(u)) \Big) du \tag{19}$$

$$\leq \int_{\mathcal{U}} \bar{\eta}_\tau(u) \max_{\zeta' \in \mathcal{Z}(t_1)} \sum_{i \in [I]} p^u[i] \Big( \mathbb{E}_{\{\eta_{i,\tau}^u\}}[J_i(t_1, x^u; \eta_{i,\tau}^u, \zeta')] + \int_{t=t_0}^{t_1} l_i(u, \zeta(u)) \Big) du.$$

Here $(\eta_{i,\tau}^u, \zeta^u)$ is the strategy pair taken at $(t_1, x^u, p^u)$.

From the induction assumption we have:

$$\max_{\zeta' \in \mathcal{Z}(t_1)} \sum_{i \in [I]} p^u[i] \mathbb{E}_{\{\eta^u_{i,\tau}\}}[J_i(t_1, x^u; \eta^u_{i,\tau}, \zeta')] \leq V(t_1, x^u, p^u) + M_1(T - t_1)\tau.$$

Incorporating this and Eq. 18 into Eq. 19 to have

$$J(t_0, x_0, p; \{\eta_{i,\tau}\}, \zeta) \leq \int_{\mathcal{U}} \bar{\eta}_\tau(u)\Big(V(t_1, x^u, p^u) + \sum_{i \in [I]} p^u[i] \int_{t=t_0}^{t_0+\tau} l_i(u, \zeta(u))\Big)du + M_1(T - t_1)\tau \tag{20}$$

$$\leq \int_{\mathcal{U}} \bar{\eta}_\tau(u)\Big(V(t_0, x_0, p) + L_1 L_2 \tau^2\Big)du + M_1(T - t_1)\tau$$

Setting $M_1 = L_1 L_2$ to have

$$J(t_0, x_0, p; \{\eta_{i,\tau}\}, \zeta) \leq V(t_0, x_0, p) + M_1(T - t_0)\tau. \tag{21}$$

Since Eq. 21 holds for all $\zeta \in \mathcal{Z}(t_0)$, we get Eq. 12. The same technique applies to the dual value. $\square$

## B. Proof of Theorem 4.1

**Theorem 4.1** (A splitting reformulation of the primal and dual DPs) The RHS of Eq. 3 can be reformulated as

$$\min_{\{u^k\}, \{\alpha_{ki}\}} \max_{\{v^k\}} \sum_{k=1}^{I} \lambda^k \Big(V(t + \tau, x^k, p^k) + \tau \mathbb{E}_{i \sim p^k}[l_i(u^k, v^k)]\Big)$$

$$\text{s.t.} \quad u^k \in \mathcal{U}, \quad x^k = \text{ODE}(x, \tau, u^k, v^k; f), \quad v^k \in \mathcal{V},$$

$$\alpha_{ki} \in [0, 1], \sum_{k=1}^{I} \alpha_{ki} = 1, \lambda^k = \sum_{i=1}^{I} \alpha_{ki} p[i], \tag{$P_1$}$$

$$p^k[i] = \frac{\alpha_{ki} p[i]}{\lambda^k}, \quad \forall i, k \in [I].$$

And the RHS of Eq. 5 can be reformulated as

$$\min_{\{v^k\}, \{\lambda^k\}, \{\hat{p}^k\}} \max_{\{u^k\}} \sum_{k=1}^{I+1} \lambda^k \Big(V^*(t + \tau, x^k, \hat{p}^k - \tau l(u^k, v^k))\Big)$$

$$\text{s.t.} \quad u^k \in \mathcal{U}, \quad v^k \in \mathcal{V}, \quad x^k = \text{ODE}(x, \tau, u^k, v^k; f), \tag{$P_2$}$$

$$\lambda^k \in [0, 1], \sum_{k=1}^{I+1} \lambda^k \hat{p}^k = \hat{p}, \sum_{k=1}^{I+1} \lambda^k = 1, \ k \in [I+1].$$

*Proof.* Recall that the primal DP is:

$$V_\tau(t_0, x_0, p) = \min_{\{\eta_i\}} \mathbb{E}_{u \sim \bar{\eta}} \left[\max_{v \in \mathcal{V}} V_\tau(t_0 + \tau, x'(u, v), p'(u)) + \tau \mathbb{E}_{i \sim p'(u)}[l_i(u, v)]\right]$$

$$= \min_{\{\eta_i\}} \int_{\mathcal{U}} \bar{\eta}(u) \max_{v \in \mathcal{V}} V_\tau(t_0 + \tau, x'(u, v), p'(u)) + \tau \mathbb{E}_{i \sim p'(u)}[l_i(u, v)]du$$

$$= \min_{\{\eta_i\}} \int_{\mathcal{U}} \bar{\eta}(u) a(u, p'(u))du, \quad \Big(a(u, p'(u)) = \max_{v \in \mathcal{V}} V_\tau(t_0 + \tau, x'(u, v), p'(u)) + \tau \mathbb{E}_{i \sim p'(u)}[l_i(u, v)]\Big)$$

$$\tag{22}$$

Now we introduce a pushforward measure $\nu$ on $\Delta(I)$ for any $E \subset \Delta(I)$: $\nu(E) = \int_{\{u:p'(u) \in E\}} \bar{\eta}(u) \, du$. Let $\eta_{p'}$ be the conditional measure on $\mathcal{U}$ for each $p'$. Then we have

$$\min_{\{\eta_i\}} \int_{\mathcal{U}} \bar{\eta}(u) a(u, p'(u))du = \min_{\nu} \int_{\Delta(I)} \min_{\eta_{p'}} \Big[\int_{p'(u)=p'} a(u, p') \, \eta_{p'}(du)\Big]\nu(dp')$$

$$= \min_{\nu} \int_{\Delta(I)} \min_{u \in \mathcal{U}} a(u, p')\nu(dp')$$

$$= \min_{\nu} \int_{\Delta(I)} \tilde{a}(p')\nu(dp').$$

This leads to the following reformulation of $V_\tau$:

$$V_\tau(t_0, x_0, p) = \min_\nu \int_{\Delta(I)} \tilde{a}(p')\nu(dp') \tag{23}$$
$$\text{s.t.} \quad \mathbb{E}_\nu[p'] = p.$$

One easily notice that the RHS of Eq. 23 computes the convexificiation of $\tilde{a}(p')$ at $p' = p$. Since convexification in $\Delta(I)$ requires at most $I$ vertices, $\nu^*$ that solves Eq. 23 is $I$-atomic. We will denote by $\{p^k\}_{k \in [I]}$ the set of "splitting" points that has non-zero probability mass according to $\nu^*$, and let $\lambda^k := \nu^*(p^k)$. Using Isaacs' condition (A4), $\arg\min_{u \in \mathcal{U}} a(u, p)$ is non-empty for any $p \in \Delta(I)$, and therefore each $p^k$ is associated with (at least) one action in $\arg\min_{u \in \mathcal{U}} a(u, p^k)$. As a result, $\{\eta_i\}$ is also concentrated on a common set of $I$ actions in $\mathcal{U}$. Specifically, denote this set by $\{u^k\}_{k \in [I]}$, we should have $\alpha_{ki} := \eta_i(u^k) = \lambda^k p^k[i]/p[i]$. Thus we reach $P_1$. The same proof technique can be applied to the dual DP to derive $P_2$. $\qquad\square$

## C. Connection between Value Convexification and Nash Equilibrium in Incomplete-Information Games

Here we explain the construction of Nash equilibrium as a consequence of value convexification. For ease of exposition, we will use examples from a simplistic setting: repeated normal-form games with one-sided information. We also walk through the computation of strategies for the informed and uninformed players for the given examples. We refer readers to (Aumann et al., 1995; De Meyer, 1996; Sorin, 2002) for more details on the theoretical development.

Consider two normal-form zero-sum payoff tables given by matrices $G_1$ and $G_2$ as shown in Eq. 24. P1 is the row player with actions $\{U, D\}$ and P2 the column player with actions $\{L, R\}$. At the beginning of the game, nature picks game $G_1$ with probability $p$ and communicates that only to P1. P2 only knows the probability $p$. Both players pick their actions and announce them simultaneously for that round without knowing the resultant payoff. This process is repeated until the end of the game, at which point the average payoff is revealed. The game can be repeated either finitely or infinitely. For conciseness, we only discuss the latter case. To align the discussion with literature on repeated games, we will consider P1 maximize, rather than minimize, the payoff. We call this game $G(p)$.

$$G_1 = \begin{array}{c} \\ U \\ D \end{array}\begin{array}{c} L \quad R \\ \begin{bmatrix} 1 & 0 \\ 0 & 0 \end{bmatrix} \end{array} \qquad G_2 = \begin{array}{c} \\ U \\ D \end{array}\begin{array}{c} L \quad R \\ \begin{bmatrix} 0 & 0 \\ 0 & 1 \end{bmatrix} \end{array} \tag{24}$$

Let us assume, for simplicity, $p = 0.5$, and that the game being played is $G_1$. Since P1 knows that $G_1$ is the game, he could play $U$ every time, as $D$ would otherwise lead to a payoff of zero. However, as the game progresses, P2 will be able to deduce that $G_1$ is the game being played, forcing her to always play $R$, which guarantees a payoff of zero. Similarly, if $G_2$ is selected, and P1 always plays $D$, P2 will eventually figure out the true game, and guarantee a payoff of zero in the remainder of the game. In this particular game, P1 can improve his expected payoff by ignoring the actual game type. Then players play a complete-information game given by the expected payoff matrix $\bar{G}(p) = \begin{bmatrix} \frac{1}{2} & 0 \\ 0 & \frac{1}{2} \end{bmatrix}$, for which the optimal strategy for P1 (resp. P2) is to play $\{U, D\}$ (resp. $\{L, R\}$) with probability 0.5, leading to an expected payoff of $\frac{1}{4}$ to P1 in each round. Notice that by playing this way, P1 conceals the information about which game is being played, i.e., the public belief $p$ is always 0.5. Thus $\frac{1}{4}$ is the value of the *non-revealing* game and the corresponding strategy is known as the *non-revealing* strategy. In the above game, the non-revealing strategy is Nash. One can easily see that in the game $(-G_1, -G_2)$, a revealing strategy of P1 will instead be Nash.

It is important to note that for some games P1 will partially reveal the type information by splitting the belief in the first round. This can be seen from the following game with two possible payoff tables in Eq. 25:

$$G_1 = \begin{array}{c} \\ P1 \end{array}\begin{array}{c} P2 \\ \begin{bmatrix} 0 & 1 & 1 & 3 \\ 0 & 1 & 0 & 3 \end{bmatrix} \end{array} \qquad G_2 = \begin{array}{c} \\ P1 \end{array}\begin{array}{c} P2 \\ \begin{bmatrix} 3 & 0 & 1 & 0 \\ 3 & 1 & 1 & 0 \end{bmatrix} \end{array} \tag{25}$$

Let $p$ be the probability that the chosen game is $G_1$. Then the non-revealing game is defined by:

$$\bar{G}(p) = P1 \begin{array}{c} P2 \\ \begin{bmatrix} 3(1-p) & p & 1 & 3p \\ 3(1-p) & 1 & (1-p) & 3p \end{bmatrix} \end{array}. \tag{26}$$

Let $U(p)$ be the value of the non-revealing game, and let $V(p)$ be the value of the original game. Theorem 3.2 in (Aumann et al., 1995) says that $V(p)$ is the concave hull of $U(p)$, i.e., for any $p \in \Delta(I)$ ($I = 2$ in this case)

$$V(p) = \text{Cav } U(p). \tag{27}$$

This is because for any $p \in \Delta(I)$ where $U(p) < \text{Cav } U(p)$, P1 can play a mixed strategy to achieve an expected payoff of Cav $U(p)$, by splitting the public belief to some $I$ vertices in $\Delta(I)$. Once this splitting is done, P1 can keep on playing non-revealing strategy to maintain Cav $U(p)$ as his expected payoff. We elaborate using the example: The value of the non-revealing game $U(p)$ is

$$U(p) = \begin{cases} 3p, & 0 \le p \le 2 - \sqrt{3} \\ 1 - p(1-p), & 2 - \sqrt{3} \le p \le \sqrt{3} - 1 \\ 3(1-p), & \sqrt{3} - 1 \le p \le 1. \end{cases} \tag{28}$$

The concavification of the value is given by:

$$\text{Cav } U(p) = \begin{cases} 3p, & 0 \le p \le 2 - \sqrt{3} \\ 6 - 3\sqrt{3}, & 2 - \sqrt{3} \le p \le \sqrt{3} - 1 \\ 3(1-p), & \sqrt{3} - 1 \le p \le 1. \end{cases} \tag{29}$$

Both $U(p)$ and $V(p)$ are visualized in Fig 5. From the figure, P1 attains maximum value $6 - 3\sqrt{3}$ at $p = 2 - \sqrt{3}$ and $p = \sqrt{3} - 1$. Therefore, P1 can play a mixed strategy to attain the maximum value by announcing a mixed strategy in such a way that the public belief $p$ is updated to either $(2 - \sqrt{3})$ or $(\sqrt{3} - 1)$ depending on the action P1 actually takes. This makes P1's strategy *partially revealing* as P2 will not be able to deduce P1's true type. Specifically, for $p = 0.5$, and if the actual game is $G_1$, P1 plays the mixed strategy for $\bar{G}(2 - \sqrt{3})$ with the probability $2 - \sqrt{3}$ and for $\bar{G}(\sqrt{3} - 1)$ with probability $\sqrt{3} - 1$; if the actual game is $G_2$, he plays the mixed strategy for $\bar{G}(2 - \sqrt{3})$ with probability $\sqrt{3} - 1$ and for $\bar{G}(\sqrt{3} - 1)$ with probability $2 - \sqrt{3}$. More generally, for any nature's distribution $p$, P1's strategy is to compute $\lambda \in \Delta(I)$ and $p_i \in \Delta(I)$ such that $\sum_{i=1}^{I} \lambda[i] u(p^i) = \text{Cav}(U(p))$ and $\sum_{i=1}^{2} \lambda_i p^i = p$. Then, given his true type $k$, he plays the maximin strategy for $\bar{G}(p^i)$ with probability $\lambda_i p_k^i / p_k$. (Gilpin & Sandholm, 2008) first discussed the nonconvex problem for solving Cav $u$.

Next, we need to derive strategy for P2. Unlike P1, P2 has to guess the true game that is being played and hedge against potential manipulation from P1. A good strategy is to play in such a way that she pays the same amount to P1 no matter the type of the game. To do so, P2 plays a game with a vector payoff that contains the amount she pays to P1 for each game types.

Consider the game in Eq. 25. By observing P1's action, P2 can keep track of the vector payoffs $(x, y)$ for each stage. If at the beginning of the game P1 chose the last row and P2 chose the last column, then the vector payoff is $(3, 0)$. All possible vector payoffs define vertices in Fig. 6. The running average of the vector payoffs (the shaded region in Fig. 6) is defined by:

$$(\xi_n, \eta_n) = \left( \frac{1}{n}(x_1 + x_2 + \cdots + x_n), \frac{1}{n}(y_1 + y_2 + \cdots + y_n) \right). \tag{}$$

P2 knows that if $G_1$ (resp. $G_2$) is the game, P1 will move the average to the right (resp. top). (Blackwell, 1956) first discussed P2's strategy to minimize the average payoff by introducing the concept of *approachability*: A set $S$ in the payoff vector space is *approachable* for P2 if P2 can adopt a strategy ensuring that the distance of the running vector payoff from $S$ converges to zero with probability one, regardless of P1's strategy.

From the primal game, we know that P1 can guarantee payoff of $6 - 3\sqrt{3}$ (the dashed lines in Fig. 7). To construct the approachable set of P2, consider P1's mixed strategy as $(\pi, 1 - \pi)$ and P2's mixed strategy as $(\sigma_1, \sigma_2, \sigma_3, \sigma_4)$. We can determine the expected payoffs to P1: When P2 plays first column, the payoff to P1 is $(0, 3)$, when she plays the second, it

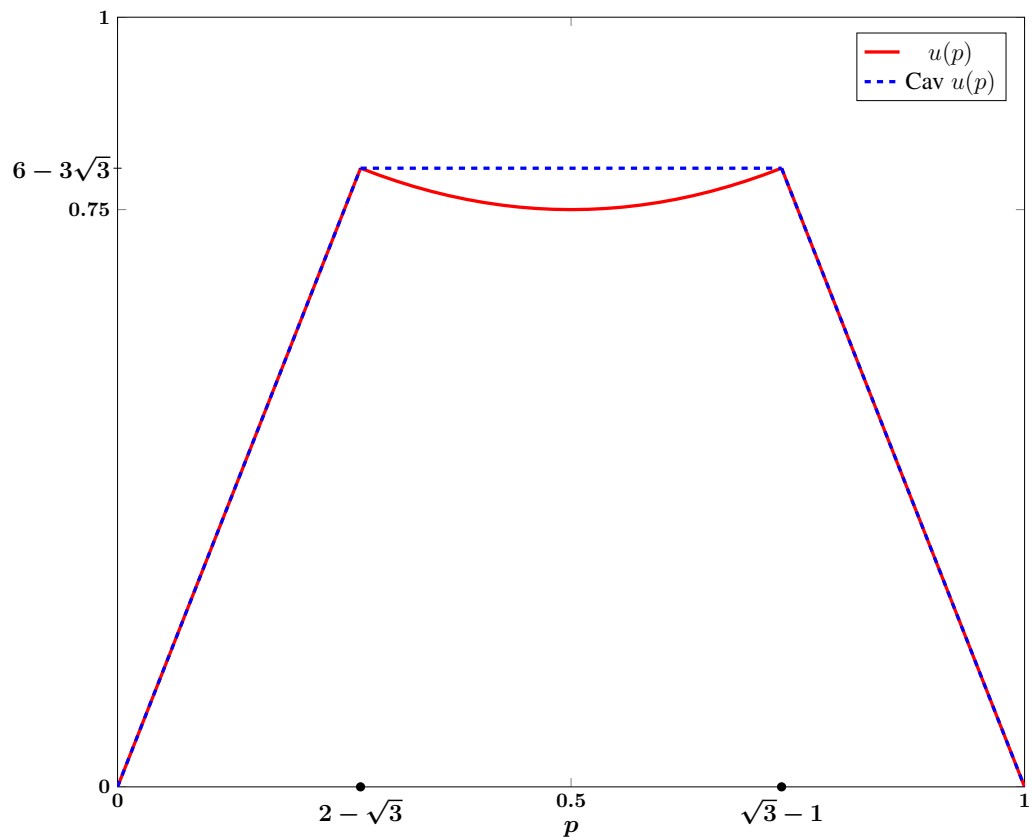

*Figure 5.* Non-revealing game value $u_1$ and its concavification

is $(1, 1 - \pi)$, and so on. Thus, for all possible $(\sigma_1, \sigma_2, \sigma_3, \sigma_4)$, the expected payoffs to P1 is the convex hull of the points $(0, 3), (1, 1 - \pi), (\pi, 1 - \pi), (3, 0)$. Denote the shaded region in Fig. 7 as $S = \{\xi_n, \eta_n : (\xi_n, \eta_n) \leq 6 - \sqrt{3}\}$.

The optimal strategy for P2 is as follows. P2 keeps track of average vector payoff (say $g_n = (\xi_n, \eta_n)$). If $g_n \in S$, then P2 plays arbitrarily. However, if $g_n \notin S$, P2 must project the vector $g_n$ onto the closest point $c = \arg \min_{m \in C} ||g_n - m||$. P2 then adopts the mixed strategy corresponding to the projection $q = (g_n - c)/||g_n - c|| \in \Delta(K)$ (here, $K = 2$), and plays optimally in the game $G(q)$.

## D. Connection between Primal and Dual Games

Here we continue to use the infinitely-repeated game setting to explain the connection between the primal and the dual games and the interpretation of the dual variable $\hat{p}$. Please see Theorem 2.2 in (De Meyer, 1996) and the extension to differential games in (Cardaliaguet, 2007).

Let the primal game be $G(p)$ for $p \in \Delta(I)$, the dual game be $G^*(\hat{p})$ for $\hat{p} \in \mathbb{R}^I$, and let $\{\eta_i\}_{i=1}^I$ be the set of strategies for P1 and $\zeta$ the strategy for P2. $\eta_i \in \Delta(d_u)$ and $\zeta \in \Delta(d_v)$. We note that P1's strategy $\{\eta_i\}_{i=1}^I$ can also be together represented in terms of $\pi := \{\pi_{ij}\}^{I, d_u}$ such that $\sum_j^{d_u} \pi_{ij} = p[i]$ and $\eta_i[j] = \pi_{ij}/p[i]$, i.e., nature's distribution is the marginal of $\pi$ and P1's strategy the conditional of $\pi$. Let $G^i_{\eta\zeta}$ be the payoff to P1 of type $i$ for strategy profile $(\eta, \zeta)$. We have the following results connecting $G(p)$ and $G^*(\hat{p})$:

1. If $\pi$ is Nash for P1 in $G(p)$ and $\hat{p} \in \partial V(p)$, then $\{\eta_i\}_{i=1}^I$ is also Nash for P1 in $G^*(\hat{p})$.

2. If $\pi$ is Nash for P1 in $G^*(\hat{p})$ and $p$ is induced by $\pi$, then $p \in \partial V^*(\hat{p})$ and $\pi$ is Nash for P1 in $G(p)$.

3. If $\zeta$ is Nash for P2 in $G^*(\hat{p})$ and $p \in \partial V^*(\hat{p})$, then $\zeta$ is also Nash for P2 in $G(p)$.

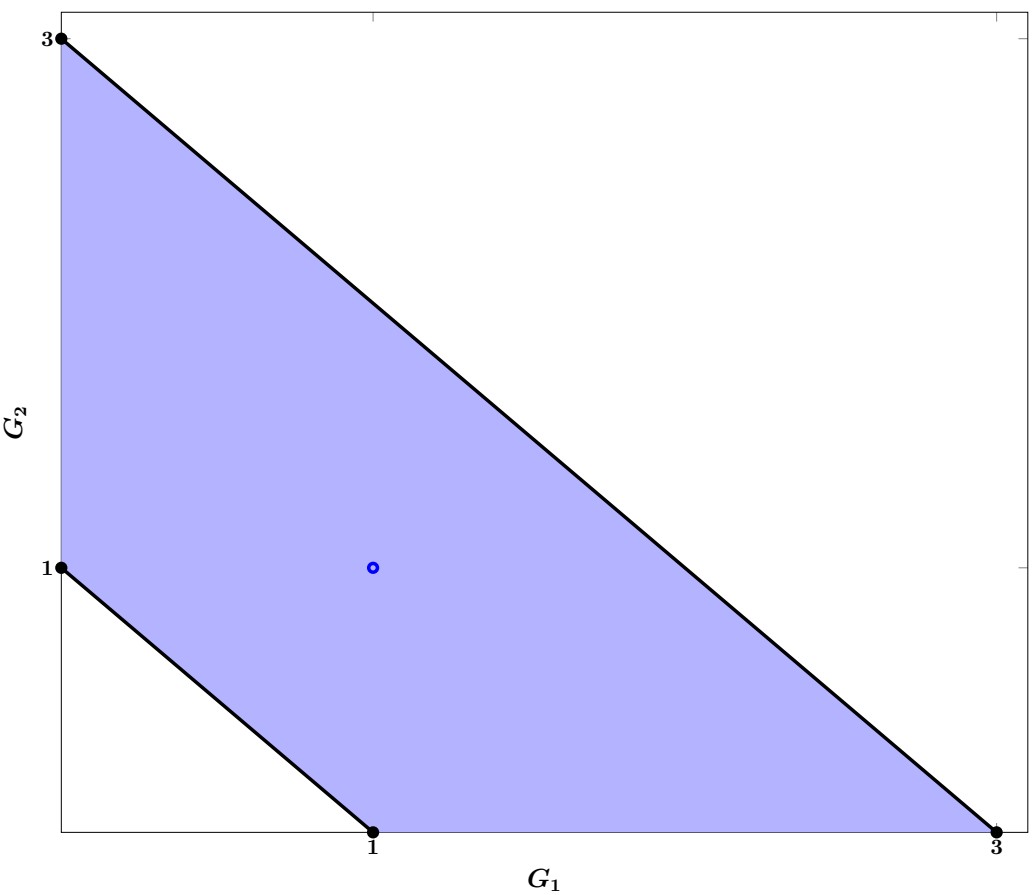

*Figure 6.* Game from P2's perspective.

4. If $\zeta$ is Nash for $G(p)$, and let $\hat{p}^i := \max_{\eta \in \Delta(d_u)} G^i_{\eta\zeta}$ and $\hat{p} := [\hat{p}^1, ..., \hat{p}^I]^T$, then $p \in \partial V^*(\hat{p})$ and $\zeta$ is also Nash for P2 in $G^*(\hat{p})$.

From the last two properties we have: If $\zeta$ is Nash for $G(p)$ and $G^*(\hat{p})$, then $\hat{p} = \max_{\eta \in \Delta(d_u)} G^i_{\eta\zeta}$, i.e., $\hat{p}[i]$ is the payoff of type $i$ if P1 plays a best response for that type to P2's Nash.

## E. Analytical Examples

The following examples are reproduced from Ghimire et al. (2024) with permission.

### E.1. Hexner's Game: Analytical Solution

Here we discuss the solution to Hexner's game using primal and dual formulations (i.e., Eq. P$_1$ and Eq. P$_2$) on a differential game as proposed in Hexner (1979). Consider two players with linear dynamics

$$\dot{x}_i = A_i x_i + B_i u_i,$$

for $i = 1, 2$, where $x_i(t) \in \mathbb{R}^{d_x}$ are system states, $u_i(t) \in \mathcal{U}$ are control inputs belonging to the admissible set $\mathcal{U}$, $A_i, B_i \in \mathbb{R}^{d_x \times d_x}$. Let $\theta \in \{-1, 1\}$ be Player 1's type unknown to Player 2. Let $p_\theta$ be Nature's probability distribution of $\theta$. Consider that the game is to be played infinite many times, the payoff is an expectation over $\theta$:

$$J(u_1, u_2) = \mathbb{E}_\theta \Big[ \int_0^T \Big( \|u_1\|^2_{R_1} - \|u_2\|^2_{R_2} \Big) dt +$$
$$\|x_1(T) - z\theta\|^2_{K_1(T)} - \|x_2(T) - z\theta\|^2_{K_2(T)} \Big], \tag{30}$$

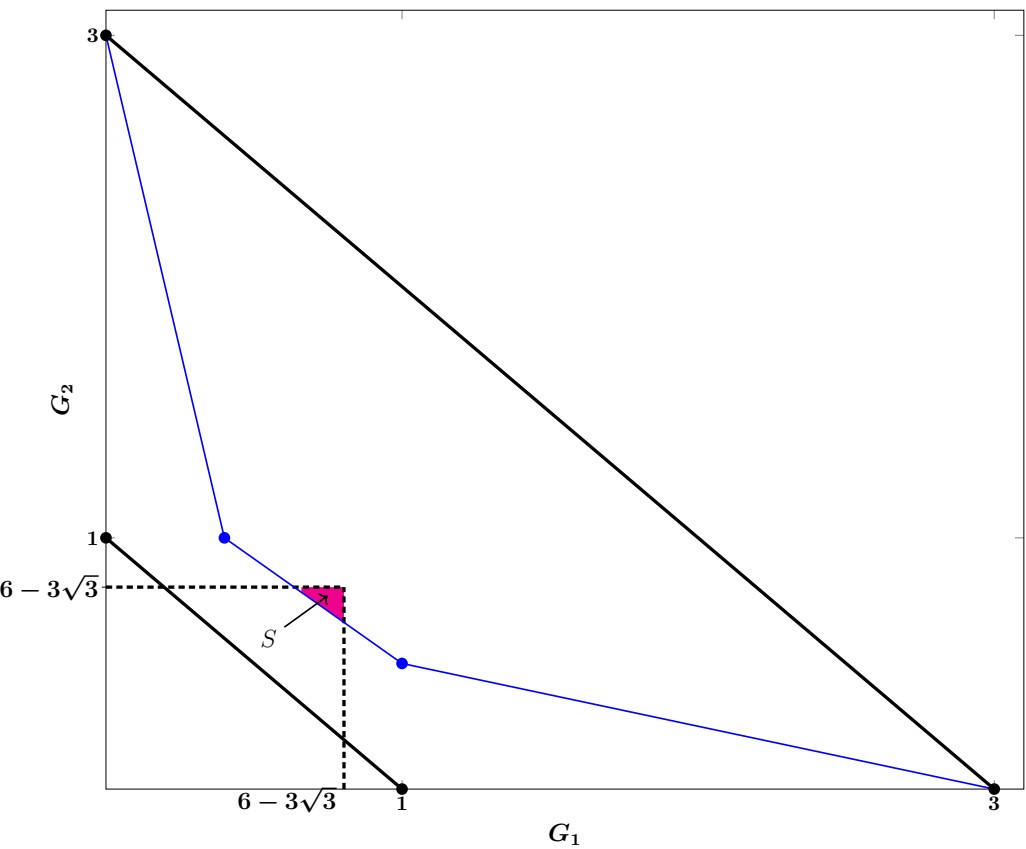

*Figure 7.* Approachable set (shaded in magenta) of P2

where, $z \in \mathbb{R}^{d_x}$. $R_1$ and $R_2$ are continuous, positive-definite matrix-valued functions, and $K_1(T)$ and $K_2(T)$ are positive semi-definite matrices. All parameters are publicly known except for $\theta$, which remains private. Player 1's objective is to get closer to the target $z\theta$ than Player 2. However, since Player 2 can deduce $\theta$ indirectly through Player 1's control actions, Player 1 may initially employ a non-revealing strategy. This involves acting as though he only knows about the prior distribution $p_\theta$ (rather than the true $\theta$) to hide the information, before eventually revealing $\theta$.

First, it can be shown that players' control has a 1D representation, denoted by $\tilde{\theta}_i \in \mathbb{R}$, through:

$$u_i = -R_i^{-1} B_i^T K_i x_i + R_i^{-1} B_i^T K_i \Phi_i z \tilde{\theta}_i,$$

for $i = 1, 2$, where $\dot{\Phi}_i = A_i \Phi_i$ with boundary condition $\Phi_i(T) = I$, and

$$\dot{K}_i = -A_i^T K_i - K_i A_i + K_i^T B_i R_i^{-1} B_i^T K_i.$$

Then define a quantity $d_i$ as:

$$d_i = z^T \Phi_i^T K_i B_i R_i^{-1} B_i^T K_i^T \Phi_i z. \tag{31}$$

With these, the game can be reformulated with the following payoff function:

$$J(t, \tilde{\theta}_1, \tilde{\theta}_2) = \mathbb{E}_\theta \left[ \int_{\tau=t}^{T} (\tilde{\theta}_1(\tau) - \theta)^2 d_1(\tau) - (\tilde{\theta}_2(\tau) - \theta)^2 d_2(\tau) d\tau \right], \tag{32}$$

where $d_1$, $d_2$, $p_\theta$ are common knowledge; $\theta$ is only known to Player 1; the scalar $\tilde{\theta}_1$ (resp. $\tilde{\theta}_2$) is Player 1's (resp. Player 2's) strategy. We consider two player types $\theta \in \{-1, 1\}$ and therefore $p_\theta \in \Delta(2)$.

Then by defining critical time:

$$t_r = \arg\min_t \int_0^t (d_1(s) - d_2(s)) ds,$$

we have the following equilibrium:

$$\tilde{\theta}_1(s) = \tilde{\theta}_2(s) = 0 \quad \forall s \in [0, t_r] \tag{33}$$

$$\tilde{\theta}_1(s) = \tilde{\theta}_2(s) = \theta \quad \forall s \in (t_r, T], \tag{34}$$

To solve the game via primal-dual formulation, we introduce a few quantities. First, introduce time stamps $[T_k]_{k=1}^{2r}$ as roots of the time-dependent function $d_1 - d_2$, with $T_0 = 0$, $T_{2q+1} = t_r$, and $T_{2r+1} = T$. Without loss of generality, assume:

$$d_1 - d_2 < 0 \quad \forall t \in (T_{2k}, T_{2k+1}) \; \forall k = 0, ..., r, \tag{35}$$

$$d_1 - d_2 \geq 0 \quad \forall t \in [T_{2k-1}, T_{2k}] \; \forall k = 1, ..., r. \tag{36}$$

Also introduce $D_k := \int_{T_k}^{T_{k+1}} (d_1 - d_2) ds$ and

$$\tilde{D}_k = \begin{cases} \tilde{D}_{k+1} + D_k, & \text{if } \tilde{D}_{k+1} + D_k < 0 \\ 0, & \text{otherwise} \end{cases}, \tag{37}$$

with $\tilde{D}_{2r+1} = 0$.

The following properties are necessary (see (Hexner, 1979) for details):

1. $\int_k^{2q+1} (d_1 - d_2) ds = \sum_k^{2q} D_k < 0, \forall k = 0, ..., 2q$;

2. $\int_{2q+1}^{k} (d_1 - d_2) ds = \sum_{2q+1}^{k-1} D_k > 0, \forall k = 2q+2, ..., 2r+1$;

3. $\tilde{D}_{2q+2} + D_{2q+1} > 0$;

4. $\tilde{D}_k < 0, \forall k < 2q+1$.

**Primal game.** We start with $V(T, p) = 0$ where $p := p_\theta[1] = \Pr(\theta = -1)$. The Hamiltonian is as follows:

$$H(p) = \min_{\tilde{\theta}_1} \max_{\tilde{\theta}_2} \mathbb{E}_\theta \left[ (\tilde{\theta}_1 - \theta)^2 d_1 - (\tilde{\theta}_2 - \theta)^2 d_2 \right]$$
$$= 4p(1-p)(d_1 - d_2).$$

The optimal actions for the Hamiltonian are $\tilde{\theta}_1 = \tilde{\theta}_2 = 1 - 2p$. From Bellman backup, we can get

$$V(T_k, p) = 4p(1-p)\tilde{D}_k.$$

Therefore, at $T_{2q+1}$, we have

$$V(T_{2q+1}, p) = Vex_p \left( V(T_{2q+2}, p) + 4p(1-p)D_{2q+1} \right)$$
$$= Vex_p \left( 4p(1-p)(\tilde{D}_{2q+2} + D_{2q+1}) \right).$$

Notice that $\tilde{D}_{2q+2} + D_{2q+1} > 0$ (property 3) and $\tilde{D}_k < 0$ for all $k < 2q+1$ (property 4), $T_{2q+1}$ is the first time such that the right-hand side term inside the convexification operator, i.e., $4p(1-p)(\tilde{D}_{2q+2} + D_{2q+1})$, becomes concave. Therefore, splitting of belief happens at $T_{2q+1}$ with $p^1 = 0$ and $p^2 = 1$. Player 1 plays $\tilde{\theta}_1 = -1$ (resp. $\tilde{\theta}_1 = 1$) with probability 1 if $\theta = -1$ (resp. $\theta = 1$), i.e., Player 1 reveals its type. This result is consistent with Hexner's.

**Dual game.** To find Player 2's strategy, we need to derive the conjugate value which follows

$$V^*(t, \hat{p}) = \begin{cases} \max_{i \in \{1,2\}} \hat{p}[i] & \forall t \geq T_{2q+1} \\ \hat{p}[2] - \tilde{D}_t \left( 1 - \frac{\hat{p}[1] - \hat{p}[2]}{4\tilde{D}_t} \right)^2 & \forall t < T_{2q+1}, \; 4\tilde{D}_t \leq \hat{p}[1] - \hat{p}[2] \leq -4\tilde{D}_t \\ \hat{p}[1] & \forall t < T_{2q+1}, \; \hat{p}[1] - \hat{p}[2] \geq 4\tilde{D}_t \\ \hat{p}[2] & \forall t < T_{2q+1}, \; \hat{p}[1] - \hat{p}[2] < 4\tilde{D}_t \end{cases}$$

Here $\hat{p} \in \nabla_{p_\theta} V(0, p_\theta)$ and $V(0, p_\theta) = 4p[1]p[2]\tilde{D}_0$. For any particular $p_* \in \Delta(2)$, from the definition of subgradient, we have $\hat{p}[1]p_*[1] + \hat{p}[2]p_*[2] = 4p_*[1]p_*[2]\tilde{D}_0$ and $\hat{p}[1] - \hat{p}[2] = 4(p_*[2] - p_*[1])\tilde{D}_0$. Solving these to get $\hat{p} = [4p_*[2]^2\tilde{D}_0, 4p_*[1]^2\tilde{D}_0]^T$. Therefore $\hat{p}[1] - \hat{p}[2] = 4\tilde{D}_0(1 - 2p_*[1]) \in [4\tilde{D}_0, -4\tilde{D}_0]$, and

$$V^*(0, \hat{p}) = \hat{p}[2] - \tilde{D}_0 \left( 1 - \frac{\hat{p}[1] - \hat{p}[2]}{4\tilde{D}_0} \right)^2 .$$

Notice that $V^*(t, \hat{p})$ is convex to $\hat{p}$ since $\tilde{D}_0 < 0$ (property 4) for all $t \in [0, T]$. Therefore, there is no splitting of $\hat{p}$ during the dual game, i.e., $\hat{\theta}_2 = 1 - 2p$. This result is also consistent with results in Hexner (1979).

### E.2. Example of a Turn-Based Game

We present a zero-sum variant of the classic beer-quiche game, which is a turn-based incomplete-information game with a perfect Bayesian equilibrium. Unlike in Hexner's game, Player 1 in beer-quiche game wants to maximize his payoff, and Player 2 wants to minimize it; hence, Vex becomes a Cav. We solve the game through backward induction (from $t = 2, 1, 0$)

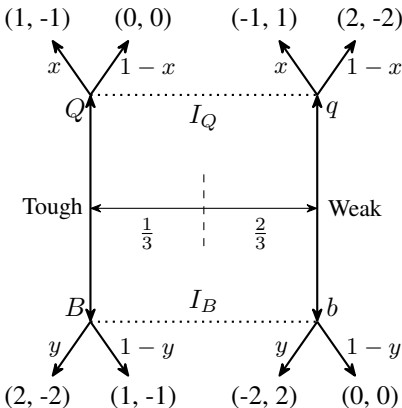

*Figure 8.* Zero-Sum Beer-Quiche Game

of its primal and dual values (denoted by $V$ and $V^*$ respectively). Players 1 and 2 make their respective decisions at $t = 0$ and $t = 1$, and the game ends at $t = 2$. The state $x$ at a time $t$ encodes the history of actions taken until $t$.

**Primal game:** First, we compute the equilibrium strategy of Player 1 using the primal value. At the terminal time step ($t = 2$), based on Fig. 8, the value for Player 1 is the following:

$$V(2, x, p) = \begin{cases} 4p_T - 2 & \text{if } x = (B, b) \\ p_T & \text{if } x = (B, d) \\ 2p_T - 1 & \text{if } x = (Q, b) \\ 2 - 2p_T & \text{if } x = (Q, d) \end{cases} . \tag{38}$$

At the intermediate time step ($t = 1$), it is Player 2's turn to take an action. Therefore, the value is a function of Player 1's action at $t = 0$ and Player 2's current action. And for the same reason, the value is not a *concavification* (Cav) over the RHS term.

$$V(1, x, p) = \min_{v \in \{b, d\}} V(2, (x, v), p). \tag{39}$$

We can find the best responses of Player 2 for both actions of Player 1. This leads to

$$V(1, x, p) = \begin{cases} p_T & \text{if } x = B, \ 3p_T - 2 \geq 0 \quad (v^* = d) \\ 4p_T - 2 & \text{if } x = B, \ 3p_T - 2 < 0 \quad (v^* = b) \\ 2 - 2p_T & \text{if } x = Q, \ 4p_T - 3 \geq 0 \quad (v^* = d) \\ 2p_T - 1 & \text{if } x = Q, \ 4p_T - 3 < 0 \quad (v^* = b) \end{cases} . \tag{40}$$

Finally, at the beginning of the game ($t = 0$), we have

$$V(0, \emptyset, p) = \text{Cav} \left( \max_{u \in \{B, Q\}} V(1, u, p) \right). \tag{41}$$

Cav is achieved by taking the concave hull with respect to $p_T$:

$$V(0, \emptyset, p) = \begin{cases} 5p_T/2 - 1 & \text{if } p_T < 2/3 \\ p_T & \text{if } p_T \geq 2/3 \end{cases}. \tag{42}$$

When $p_T \in [0, 2/3)$,

$$V(0, \emptyset, p) = \lambda \max_u V(1, u, p^1) + (1 - \lambda) \max_u V(1, u, p^2),$$

where $p^1 = [0, 1]^T$, $p^2 = [2/3, 1/3]^T$, and $\lambda p^1 + (1 - \lambda)p^2 = p$.

Therefore, when $p_T = 1/3$, $\lambda = 1/2$, Player 1's strategy is:

$$\Pr(u = Q|T) = \frac{\lambda p^1[1]}{p[1]} = 0, \qquad \Pr(u = Q|W) = \frac{\lambda p^1[2]}{p[2]} = 3/4,$$

$$\Pr(u = B|T) = \frac{(1 - \lambda)p^2[1]}{p[1]} = 1, \quad \Pr(u = B|W) = \frac{(1 - \lambda)p^2[2]}{p[2]} = 1/4. \tag{43}$$

**Dual game:** To solve the equilibrium of Player 2, we first derive the dual variable $\hat{p} \in \partial_p V(0, \emptyset, p)$ for $p = [1/3, 2/3]^T$. By definition, $\hat{p}^T p$ defines the concave hull of $V(0, \emptyset, p)$, and therefore we have

$$[1/3, 2/3]\hat{p} = V(0, \emptyset, p) = -1/6$$
$$[0, 1]\hat{p} = V(0, \emptyset, [0, 1]) = -1. \tag{44}$$

This leads to $\hat{p} = [3/2, -1]^T$.

At the terminal time, we have

$$V^*(2, x, \hat{p}) = \min\{\hat{p}[1] - g_T(x), \hat{p}[2] - g_W(x)\}$$

$$= \begin{cases} \min\{\hat{p}[1] - 2, \hat{p}[2] + 2\} & \text{if } x = (B, b) \\ \min\{\hat{p}[1] - 1, \hat{p}[2]\} & \text{if } x = (B, d) \\ \min\{\hat{p}[1] - 1, \hat{p}[2] + 1\} & \text{if } x = (Q, b) \\ \min\{\hat{p}[1], \hat{p}[2] - 2\} & \text{if } x = (Q, d) \end{cases} \tag{45}$$

At $t = 1$, we have

$$V^*(1, u, \hat{p}) = \text{Cav}_{\hat{p}} \left( \max_v V^*(2, (u, v), \hat{p}) \right). \tag{46}$$

When $u = B$, the conjugate value is a concave hull of a piece-wise linear function:

$$V^*(1, B, \hat{p}) = \text{Cav}_{\hat{p}} \left( \begin{cases} \hat{p}[1] - 1 & \text{if } \hat{p}[2] \geq \hat{p}[1] - 1 & (v^* = d) \\ \hat{p}[2] & \text{if } \hat{p}[2] \in [\hat{p}[1] - 2, \hat{p}[1] - 1) & (v^* = b) \\ \hat{p}[1] - 2 & \text{if } \hat{p}[2] \in [\hat{p}[1] - 4, \hat{p}[1] - 2) & (v^* = d) \\ \hat{p}[2] + 2 & \text{if } \hat{p}[2] < \hat{p}[1] - 4 & (v^* = b) \end{cases} \right)$$

$$= \begin{cases} \hat{p}[1] - 1 & \text{if } \hat{p}[2] \geq \hat{p}[1] - 1 & (v^* = d) \\ 2/3\hat{p}[1] + 1/3\hat{p}[2] - 2/3 & \text{if } \hat{p}[2] \in [\hat{p}[1] - 4, \hat{p}[1] - 1) & (\text{mixed strategy}) \\ \hat{p}[2] + 2 & \text{if } \hat{p}[2] < \hat{p}[1] - 4 & (v* = b) \end{cases} \tag{47}$$

For $\hat{p} = [3/2, -1]^T$ which satisfies $\hat{p}[2] \in [\hat{p}[1] - 4, \hat{p}[1] - 1)$, Player 2 follows a mixed strategy determined based on $\{\lambda_1, \lambda_2, \lambda_3\} \in \Delta(3)$ and $\hat{p}^j \in \mathbb{R}^2$ for $j = 1, 2, 3$ such that:

(i) At least one of $\hat{p}^j$ for $j = 1, 2, 3$ should satisfy $\hat{p}[2] = \hat{p}[1] - 1$ and another $\hat{p}[2] = \hat{p}[1] - 4$. These conditions are necessary for $V^*(1, B, \hat{p})$ to be a concave hull:

$$V^*(1, B, \hat{p}) = \sum_{j=1}^3 \lambda_j \max_v V^*(2, (B, v), \hat{p}^j). \tag{48}$$

(ii) $\sum_{j=1}^3 \lambda_j \hat{p}^j = \hat{p}$.

These conditions lead to $\lambda_1 = 1/2$ and $\lambda_2 + \lambda_3 = 1/2$. Therefore, when Player 1 picks beer, Player 2 chooses to defer and bully with equal probability.

When $u = Q$, we similarly have

$$V^*(1, Q, \hat{p}) = \begin{cases} \hat{p}[1] & \text{if } \hat{p}[2] \geq \hat{p}[1] + 2 & (v^* = d) \\ ... & \text{if } \hat{p}[2] \in [\hat{p}[1] - 2, \hat{p}[1] + 2) & \text{(mixed strategy)} \\ \hat{p}[2] + 1 & \text{if } \hat{p}[2] < \hat{p}[1] - 2 & (v* = b) \end{cases} \tag{49}$$

The derivation of the concave hull when $\hat{p}[2] \in [\hat{p}[1] - 2, \hat{p}[1] + 2)$ is omitted, because, for $\hat{p} = [3/2, -1]^T$, $V^*(1, Q, \hat{p}) = \hat{p}[2] + 1 = 0$ while $v^* = b$, i.e. if Player 1 picks quiche, Player 2 chooses to bully with a probability of 1.

# F. Computational Complexity of Existing Algorithms for Solving 2p0s Normal Form Games

Here we reveal the computational complexity (in terms of the number of iterations) of some important existing algorithms for solving 2p0s normal form games. The purpose is to show that these algorithms all scale with the action space size, which limits them from solving games with continuous action spaces with discretization leads to undesirable solutions. We omit discussions about IIEFGs since they can be reformulated as NFGs.

Consider the following minimax formulation for NFGs:

$$\min_{x \in \Delta(I)} \max_{y \in \Delta(J)} x^T A y + \alpha g_1(x) - \alpha g_2(y), \tag{50}$$

where $I$ and $J$ are positive integers, $A \in \mathbb{R}^{I \times J}$ is a payoff matrix, and $g_1$, $g_2$ are strictly convex functions (e.g., L2 norm, negative entropy for NFGs, and dilated entropy for EFGs). Since Eq. 50 is convex to $x$ and concave to $y$, there exists a unique solution. When $\alpha = 0$, the solution $(x^*, y^*)$ is a Nash equilibrium, otherwise if $\alpha > 0$, the solution is an quantal response equilibrium (QRE).

**Counterfactual regret minimization.** CFR variants are average-time convergent algorithms for solving NFGs and EFGs, leveraging the fact that minimizing counterfactual regrets at all infostates achieves Nash for 2p0s games (Zinkevich et al., 2007). **Algorithm:** Here we introduce the standard CFR and CFR+. For simplicity, we will focus on solving the NFG in Eq. 50 with $\alpha = 0$ (which reduces CFR to regret matching and CFR+ to regret matching+). Given strategy profile $(x_t, y_t)$ at iteration $t \in [T]$, the instantaneous regret vector for Player 1 (resp. Player 2) is $r_1^t = A y_t - x_t^T A y_t$ (resp. $r_2^t = A^T x_t - x_t^T A y_t$). The non-negative regret vector is $R_i^t = \max\{\sum_{\tau=1}^t r_i^\tau, 0\}$ for $i \in [2]$. CFR updates the strategies as

$$x_{t+1} = \frac{R_1^t}{< \mathbf{1}, R_1^t >}, \quad y_{t+1} = \frac{R_2^t}{< \mathbf{1}, R_2^t >} \tag{51}$$

if the sums $< \mathbf{1}, R_i^t >$ is positive. Otherwise the strategy is updated as $x_{t+1} = \mathbf{1}/I$ for Player 1 and $y_{t+1} = \mathbf{1}/J$ for Player 2. CFR+ is different from CFR only in the definition of the instantaneous regret: $\hat{r}_i^t = \max\{r_i^t, 0\}$ and then $R_i^t = \max\{\sum_{\tau=1}^t \hat{r}_i^\tau, 0\}$. **Complexity:** To reach $\varepsilon$-Nash, the best-known upper bound on the complexity of CFR and CFR+ is $\mathcal{O}((I + J)/\varepsilon^2)$ (Cesa-Bianchi & Lugosi, 2006). While this sublinear convergence rate seems to be worse than regularized descent-ascent algorithms with guaranteed linear convergence (e.g., MMD and regularized FTRLs), CFR+ still enjoys the state-of-the-art empirical performance for a variety of large IIEFGs (Tammelin, 2014). Nonetheless, it should be noted that the complexity of CFR variants scales linearly with respect to the size of the action space.

**Magnetic mirror descent.** MMD is an extension of projected gradient descent ascent that has linear last-iterate convergence to $\alpha$-QRE for $\alpha > 0$. For ease of exposition, we set $g_1(x) = \frac{1}{2}\|x\|_2^2$ and $g_2$ is similarly defined [5] **Algorithm:** Let $\eta > 0$ be a learning rate, $(x', y') \in \text{int } \Delta(I) \times \Delta(J)$ be a "magnet". Then starting from $(x_1, y_1) \in \text{int } \Delta(I) \times \Delta(J)$, at each iteration $t \in [T]$ do

$$
\begin{aligned}
x_{t+1} &= \arg\min_{x \in \Delta(I)} x^T A y_t + \frac{\alpha}{2}\|x - x'\|_2^2 + \frac{1}{2\eta}\|x - x_t\|_2^2, \\
y_{t+1} &= \arg\min_{y \in \Delta(J)} -x_t^T A y + \frac{\alpha}{2}\|y - y'\|_2^2 + \frac{1}{2\eta}\|y - y_t\|_2^2.
\end{aligned}
\tag{52}
$$

**Complexity:** (Theorem 3.4 and Corollary 3.5 of (Sokota et al., 2022)) Let the squared error be $\varepsilon := \frac{1}{2}(\|x - x^*\|_2^2 + \|y - y^*\|_2^2)$. If $(x_t, y_t) \in \text{int } \Delta(I) \times \Delta(J)$ for all $t \in [T]$, and if $\eta$ is sufficiently small [6], then for an error threshold $\varepsilon_0 > 0$, $\varepsilon \leq \varepsilon_0$ if $T \geq \frac{\ln((I+J)/\varepsilon_0)}{\ln(1+\eta\alpha)}$. Thus MMD has complexity $\mathcal{O}(\ln((I+J)/\epsilon_0))$ with respect to the action space. **Remarks:** When $\alpha = 0$, MMD reduces to projected gradient descent ascent which is known to diverge or cycle for any positive learning rate. (Sokota et al., 2022) showed empirically that MMD can be used to solve Nash by either annealing the amount of regularization over time or by having the magnet trail behind the current iterate. However, it is important to note that MMD assumes the solution to be interior, which is not the case in the games we consider when value is convex (no splitting) due to Isaacs' condition.

**FTRL variants.** FTRL is a classic online learning algorithm known to converge in potential games but cycle in Hamiltonian games (Heliou et al., 2017; Mertikopoulos et al., 2018; Liu et al., 2024). To this end, variants of FTRL have been proposed to achieve last-iterate convergence to $\epsilon$-Nash or $\epsilon$-QRE (Perolat et al., 2021). Below we introduce a few of them to show that their complexities all increase with the size of the action space. **Algorithm:** *RegFTRL* (Liu et al., 2024) introduces regularization terms $(\phi_1, \phi_2)$ that are strictly convex and continuously differentiable on their respective simplex. For each iteration, do

$$
\begin{aligned}
x_{t+1} &= \arg\min_{x \in \Delta(I)} <x, \bar{y}_t> + \phi_1(x), & \bar{y}_t &= \sum_{\tau=1}^{t} A y_\tau + \alpha \nabla g_1(x_\tau), \\
y_{t+1} &= \arg\min_{y \in \Delta(J)} - <\bar{x}_t, y> + \phi_2(y), & \bar{x}_t &= \sum_{\tau=1}^{t} A^T x_\tau + \alpha \nabla g_2(y_\tau).
\end{aligned}
\tag{53}
$$

**Complexity:** *RegFTRL* is guaranteed to find an $\varepsilon$-QRE in $\mathcal{O}\left(\frac{\ln((I+J)/\varepsilon)}{\ln(1+\eta\alpha)}\right)$ iterations (Theorem 2 in (Liu et al., 2024)). *FTRL-SP* (Abe et al., 2024) and *OMWU* (Rakhlin & Sridharan, 2013; Syrgkanis et al., 2015) finds $\varepsilon$-QRE in $\mathcal{O}\left(\frac{\ln((I+J)/\varepsilon)}{-\ln(1-\eta\alpha/2)}\right)$.

# G. Prediction Error of Value Approximation

Here we show that the baseline algorithm (Alg. 1) shares the same exponential error propagation as in standard approximate value iteration (AVI). The only difference is that the measurement error in Alg. 1 comes from numerical approximation of the minimax problems rather than randomness in state transition and rewards as in AVI. To start, let the true value be $V(t, x, p)$. Following (Zanette et al., 2019), the prediction error $\epsilon_t^{bias} := \max_{x,p} |\hat{V}_t(x, p) - V(t, x, p)|$ is affected by (1) the prediction error $\epsilon_{t+\tau}^{bias}$ propagated back from $t + \tau$, (2) the minimax error $\epsilon_t^{minmax}$ caused by limited iterations in solving the minimax problem at each collocation point: $\epsilon_t^{minmax} = \max_{(x,p) \in \mathcal{S}_t} |\tilde{V}(t, x, p) - V(t, x, p)|$, and (3) the approximation error due to the fact that $V(t, \cdot, \cdot)$ may not lie in the model hypothesis space $\mathcal{V}_t$ of $\hat{V}_t$: $\epsilon_t^{app} = \max_{x,p} \min_{\hat{V}_t \in \mathcal{V}_t} |\hat{V}_t(x, p) - V(t, x, p)|$.

**Approximation error.** For simplicity, we will abuse the notation by using $x$ in place of $(x, p)$ and omit time dependence of variables when possible. In practice we consider $\hat{V}_t$ as neural networks that share the architecture and the hypothesis space. Note that $\hat{V}_T(\cdot) = V(T, \cdot)$ is analytically defined by the boundary condition and thus $\epsilon_T^{app} = \epsilon_T^{bias} = 0$. To enable the analysis on neural networks, we adopt the assumption that $\hat{V}$ is infinitely wide and that the resultant neural tangent kernel (NTK) is positive definite. Therefore from NTK analysis (Jacot et al., 2018), $\hat{V}$ can be considered a kernel machine equipped with a kernel function $r(x^{(i)}, x^{(j)}) := <\phi(x^{(i)}), \phi(x^{(j)})>$ defined by a feature vector $\phi : \mathcal{X} \to \mathbb{R}^{d_\phi}$. Given training data $\mathcal{S} = \{(x^{(i)}, V^{(i)})\}$, let $r(x)[i] := r(x^{(i)}, x^{(j)})$, $R_{ij} := r(x^{(i)}, x^{(j)})$, $V_{\mathcal{S}} := [V^{(1)}, ..., V^{(N)}]^\top$,

---

[5]In (Sokota et al., 2022), the authors used a more general regularization definition by introducing the Bregman divergence.

[6]See Corollary D.6 in (Sokota et al., 2022) for details on the bound of $\eta$.

$\Phi_{\mathcal{S}} := [\phi(x^{(1)}), ..., \phi(x^{(N)})]$, and $w_{\mathcal{S}} := \Phi_{\mathcal{S}}(\Phi_{\mathcal{S}}^{\top}\Phi_{\mathcal{S}})^{-1}V_{\mathcal{S}}$ be model parameters learned from $\mathcal{S}$, then

$$\hat{V}(x) = r(x)^{\top}R^{-1}V_{\mathcal{S}} = <\phi(x), w_{\mathcal{S}}> \tag{54}$$

is a linear model in the feature space. Let $\theta^{\phi(x)} := r(x)^{\top}R^{-1}$ and $C := \max_x \|\theta^{\phi(x)}\|_1$. Further, let $\mathcal{S}^{\dagger} := \arg\min_{\mathcal{S}} | < \phi(x), w_{\mathcal{S}} > -V(x)|$ and $w^{\dagger} := w_{\mathcal{S}^{\dagger}}$, i.e., $w^{\dagger}$ represents the best hypothetical model given sample size $N$. Since $N$ is finite, the data-dependent hypothesis space induces an approximation error $\epsilon_t^{app} := \max_x | < \phi(x), w^{\dagger} > -V(x)|$. From standard RKHS analysis, we have $\epsilon_t^{app} \propto N^{-1/2}$.

**Error propagation.** Recall that we approximately solve $\mathrm{P}_1$ at each collocation point. Let $z := \{\lambda, p, u, v\}$ be the collection of variables and $\tilde{z}$ be the approximated saddle point resulting from DS-GDA. Let $\tilde{V}(t, x, \tilde{z})$ be the approximate value at $(t, x)$ and let $V(t, x, z^*)$ be the value at the true saddle point $z^*$. Lemma G.1 bounds the error of $\tilde{V}(t, x, \tilde{z})$:

**Lemma G.1.** $\max_x |\tilde{V}(t, x, \tilde{z}) - V(t, x, z^*)| \le \epsilon_{t+\tau}^{bias} + \epsilon_t^{minmax}$.

*Proof.* Note that $\sum_{k=1}^I \lambda^k = 1$. Then

$$\max_x |\tilde{V}(t, x, \tilde{z}) - V(t, x, z^*)| \le \max_x |\tilde{V}(t, x, \tilde{z}) - \tilde{V}(t, x, z^*)| + \max_x |\tilde{V}(t, x, z^*) - V(t, x, z^*)|$$

$$\le \epsilon_t^{minmax} + \max_x |\sum_{k=1}^I \lambda^k(\tilde{V}(t+\tau, x', p^k) - V(t+\tau, x', p^k))| \tag{55}$$

$$\le \epsilon_t^{minmax} + \epsilon_{t+\tau}^{bias}.$$

$\square$

Now we can combine this measurement error with the inherent approximation error $\epsilon_t^{app}$ to reach the following bound on the prediction error $\epsilon_t^{bias}$:

**Lemma G.2.** $\max_x |\hat{V}_t(x) - V(t, x)| \le C_t(\epsilon_t^{minmax} + \epsilon_{t+\tau}^{bias} + \epsilon_t^{app}) + \epsilon_t^{app}$.

*Proof.*

$$\max_x |\hat{V}_t(x) - V(t, x)| \le \max_x |\hat{V}_t(x) - <\phi(x), w^{\dagger}>| + \max_x | <\phi(x), w^{\dagger}> -V(t, x)|$$

$$\le \max_x | < \theta^{\phi(x)}, \tilde{V}(t, x) - V(t, x) > | + \max_x | < \theta^{\phi(x)}, V(t, x) - \phi(x)^{\top}w^{\dagger} > | + \epsilon_t^{app} \tag{56}$$

$$\le C(\epsilon_t^{minmax} + \epsilon_{t+\tau}^{bias} + \epsilon_t^{app}) + \epsilon_t^{app}.$$

$\square$

Lem. G.3 characterizes the propagation of error:

**Lemma G.3.** *Let $\epsilon_t^{app} \le \epsilon^{app}$, $\epsilon_t^{minmax} \le \epsilon^{minmax}$, and $C_t \le C$ for all $t \in [T]$. If $\epsilon_T^{app} = 0$, then $\epsilon_0^{bias} \le TC^T(\epsilon^{app} + C(\epsilon^{minmax} + \epsilon^{app}))$.*

*Proof.* Using Lem. G.2 and by induction, we have

$$\epsilon_0^{bias} \le (\epsilon^{app} + C(\epsilon^{minmax} + \epsilon^{app}))\frac{1 - C^T}{1 - C} \le TC^T(\epsilon^{app} + C(\epsilon^{minmax} + \epsilon^{app})). \tag{57}$$

$\square$

We can now characterize the computational complexity of the baseline algorithm through Thm. G.4, by taking into account the number of DS-GDA iterations and the per-iteration complexity:

**Theorem G.4.** *For a fixed $T$ and some error threshold $\delta > 0$, with a computational complexity of at least $\mathcal{O}(T^3C^{2T}I^2\epsilon^{-4}\delta^{-2})$, Alg. 1 achieves*

$$\max_{(x,p)\in\mathcal{X}\times\Delta(I)} |\hat{V}_0(x, p) - V(0, x, p)| \le \delta. \tag{58}$$

*Proof.* From Lem. G.3 and using the fact that $\epsilon^{app} \propto N^{-1/2}$, achieving a prediction error of $\delta$ at $t = 0$ requires $N = \mathcal{O}(C^{2T}T^2\delta^{-2})$. Alg. 1 solves $TN$ minimax problems, each requires a worst-case $\mathcal{O}(\epsilon^{-4})$ iterations, and each iteration requires computing gradients of dimension $\mathcal{O}(I^2)$, considering the dimensionalities of action spaces as constants. This leads to a total complexity of $\mathcal{O}(T^3C^{2T}I^2\epsilon^{-4}\delta^{-2})$. $\qquad\square$

# H. Game Settings, Baselines and Ground Truth

## H.1. Game Settings

The players move in an arena bounded between $[-1, 1]$ in all directions. All games in the paper follow 2D/3D point dynamics as follows: $\dot{x}_j = Ax_j + Bu_j$, where $x_j$ is a vector of position and velocity and $u_j$ is the action for player $j$. Note that we use $u$ and $v$ in the optimization problems P$_1$ and P$_2$ to represent player 1 and player 2's actions respectively. The type independent effort loss for each player $j$ is defined as $l_j(u_j) = u_j^\top R_j u_j$, where $R_1 = \mathtt{diag}(0.05, 0.025)$ and $R_2 = \mathtt{diag}(0.05, 0.1)$. For the higher dimensional case, $R_1 = \mathtt{diag}(0.05, 0.05, 0.025)$ and $R_2 = \mathtt{diag}(0.05, 0.05, 0.1)$.

## H.2. Ground Truth for Hexner's Game

For the 4-stage and 10-stage Hexner's game, there exists analytical solution to the equilibrium policies via solving the HJB for respective players.

$$u_j = -R_j^{-1}B_j^\top K_j x_j + R_j^{-1}B_j^\top K_j \Phi_j z\tilde{\theta}_j,$$

based on the reformulation outlined below in which players' action $\tilde{\theta}_j \in \mathbb{R}$ become 1D and are decoupled from the state: where $\Phi_j$ is a state-transition matrix that solves $\dot{\Phi}_j = A_j\Phi_j$, with $\Phi_j(T)$ being an identity matrix, and $K_j$ is a solution to a continuous-time differential Ricatti equation:

$$\dot{K}_j = -A_j^\top K_j - K_j A_j + K_j^\top B_j R_j^{-1} B_j^\top K_j, \tag{59}$$

Finally, by defining

$$d_j = z^\top \Phi_j^\top K_j B_j R_j^{-1} B_j^\top K_j^\top \Phi_i z$$

and the critical time

$$t_r = \arg\min_t \int_0^t (d_1(s) - d_2(s))ds$$

and

$$\tilde{\theta}_j(t) = \begin{cases} 0, & t \in [0, t_r] \\ \theta, & t \in (t_r, T] \end{cases}.$$

As explained in Sec.6, P1 chooses $\theta_1 = 0$ until the critical time $t_r$ and P2 follows.

Note that in order to compute the ground truth when time is discretized with some $\tau$, we need the discrete counterpart of equation 59, namely the discrete-time Ricatti difference equation and compute the matrices $K$ recursively.

## H.3. OpenSpiel Implementations and Hyperparameters

We use OpenSpiel (Lanctot et al., 2019), a collection of various environments and algorithms for solving single and multi-agent games. We select OpenSpiel due to its ease of access and availability of wide range of algorithms. The first step is to write the game environment with simultaneous moves for the stage-game and the multi-stage games (with 4 decision nodes). Note that to learn the policy, the algorithms in OpenSpiel require conversion from simultaneous to sequential game, which can be done with a built-in method.

In the single-stage game, P1 has two information states representing his type, and P2 has only one information state (i.e., the starting position of the game which is fixed). In the case of the 4-stage game, the information state (or infostate) is a vector consisting of the P1's type (2-D: [0, 1] for type-1, [1, 0] for type-2), states of the players (8-D) and actions of the players at each time step $(4 \times 2 \times U)$. The 2-D "type" vector for P2 is populated with 0 as she has no access to P1's type. For

example, the infostate at the final decision node for a type-1 P1 could be $[0, 1, x^{(8)}, \mathbb{1}_{u_0}^{(U)}, \mathbb{1}_{d_0}^{(U)}, \cdots, \mathbb{1}_{d_2}^{(U)}, \mathbf{0}^{(U)}, \mathbf{0}^{(U)}]$, and $[0, 0, x^{(8)}, \mathbb{1}_{u_0}^{(U)}, \mathbb{1}_{d_0}^{(U)}, \cdots, \mathbb{1}_{d_2}^{(U)}, \mathbf{0}^{(U)}, \mathbf{0}^{(U)}]$ for P2, where $u_k, d_k$ represent the index of the actions at $k^{th}$ decision node, $k = 0, 1, 2, 3$

The hyperparameters for DeepCFR is listed in table 3

*Table 3.* Hyperparameters for DeepCFR Training

| | |
|---|---|
| Policy Network Layers | (256, 256) |
| Advantage Network Layers | (256, 256) |
| Number of Iterations | 1000 (100, for $U = 16$) |
| Number of Traversals | 5 (10, for $U = 16$) |
| Learning Rate | 1e-3 |
| Advantage Network Batch Size | 1024 |
| Policy Network Batch Size | 10000 (5000 for $U = 16$) |
| Memory Capacity | 1e7 (1e5 for $U = 16$) |
| Advantage Network Train Steps | 1000 |
| Policy Network Train Steps | 5000 |
| Re-initialize Advantage Networks | True |

## H.4. Joint-Perturbation Simultaneous Pseudo-Gradient (JPSPG)

The core idea in the JPSPG algorithm is the use of pseudo-gradient instead of computing the actual gradient of the utility to update players' strategies. By perturbing the parameters of a utility function (which consists of the strategy), an unbiased estimator of the gradient of a smoothed version of the original utility function is obtained. Computing pseudo-gradient can often be cheaper as faster than computing exact gradient, and at the same time suitable in scenarios where the utility (or objective) functions are "black-box" or unknown. Building on top of pseudo-gradient, Martin & Sandholm (2024) proposed a method that estimates the pseudo-gradient with respect to all players' strategies simultaneously. The implication of this is that instead of multiple calls to estimate the pseudo-gradient, we can estimate the pseudo-gradient in a single evaluation. More formally, let $\mathbf{f} : \mathbb{R}^d \to \mathbb{R}^n$ be a vector-valued function. Then, its smoothed version is defined as:

$$\mathbf{f}_\sigma(\mathbf{x}) = \mathbb{E}_{\mathbf{z} \sim \mu} \mathbf{f}(\mathbf{x} + \sigma \mathbf{z}), \tag{60}$$

where $\mu$ is a $d$-dimensional standard normal distribution, $\sigma \neq 0 \in \mathbb{R}$ is a scalar. Then, extending the pseudo-gradient of a scalar-valued function to a vector-valued function, we have the following pseudo-Jacobian:

$$\nabla \mathbf{f}_\sigma(\mathbf{x}) = \mathbb{E}_{\mathbf{z} \sim \mu} \frac{1}{\sigma} \mathbf{f}(\mathbf{x} + \sigma \mathbf{z}) \otimes \mathbf{z}, \tag{61}$$

where $\otimes$ is the tensor product.

Typically, in a game, the utility function returns utility for each player given their strategy. Let $\mathbf{u} : \mathbb{R}^{n \times d} \to \mathbb{R}^n$ be the utility function in a game with $n$ players, where each player has a $d$-dimensional strategy. Then, the simultaneous gradient of $\mathbf{u}$ would be a function $\mathbf{v} : \mathbb{R}^{n \times d} \to \mathbb{R}^{n \times d}$. That is, row $i$ of $\mathbf{v}(\mathbf{u})$ is the gradient of the utility of the player $i$ with respect to its strategy, $\mathbf{v}_i = \nabla_i \mathbf{u}_i$. As a result, we can rewrite $\mathbf{v}$ concisely as: $\mathbf{v} = \text{diag}(\nabla \mathbf{u})$, where $\nabla$ is the Jacobian. With these we

have the following:

$$\mathbf{v}_\sigma(\mathbf{x}) = \mathtt{diag}(\nabla \mathbf{u}_\sigma(\mathbf{x}))$$

$$= \mathtt{diag}\left(\mathbb{E}_{\mathbf{z}\sim\mu}\frac{1}{\sigma}\mathbf{u}_\sigma(\mathbf{x}+\sigma\mathbf{z})\otimes\mathbf{z}\right)$$

$$= \mathbb{E}_{\mathbf{z}\sim\mu}\frac{1}{\sigma}\mathtt{diag}\left(\mathbf{u}_\sigma(\mathbf{x}+\sigma\mathbf{z})\otimes\mathbf{z}\right)$$

$$= \mathbb{E}_{\mathbf{z}\sim\mu}\frac{1}{\sigma}\mathbf{u}_\sigma(\mathbf{x}+\sigma\mathbf{z})\odot\mathbf{z},$$

(62)

where $\odot$ is element-wise product and a result of the fact that $\mathtt{diag}(\mathbf{a}\otimes\mathbf{b}) = \mathbf{a}\odot\mathbf{b}$. Hence, by evaluating Eq. 62 once, we get the pseudo-gradient associated with all players, making the evaluation constant as opposed to linear in number of players.

Once the pseudo-gradients are evaluated, the players update their strategy in the direction of the pseudo-gradient, assuming each player is interested in maximizing their respective utility.

**JPSPG Implementation.** In games with discrete-action spaces, where strategy is the probability distribution over the actions, JPSPG can be directly applied to get mixed strategy. However, for continuous-action games, a standard implementation would result in pure strategy solution than mixed. In order to compute a mixed strategy, we can turn into neural network as a strategy with an added randomness that can be learned as described in Martin & Sandholm (2023; 2024). We similarly define two strategy networks for each player, the outputs of which are scaled based on the respective action bounds with the help of hyperbolic tangent (tanh) activation on the final layer. The input to the strategy networks (a single hidden layered neural network with 64 neurons and output neuron of action-space dimension) are the state of the player and a random variable whose mean and variance are trainable parameters. We follow the architecture as outlined by Martin & Sandholm (2024) in their implementation of continuous-action Goofspiel. We would like to thank the authors for providing an example implementation of JPSPG on a normal-form game.

In the normal-form Hexner's game, P1's state $\mathbf{x}_1 = \{x_1, y_1, \mathtt{type}\}$, and P2's state $\mathbf{x}_2 = \{x_2, y_2\}$. $x_i$, and $y_i$ denote the x-y coordinates of the player $i$. In 4-stage case, we also include x-y velocities in the state and append the history of actions chosen by both P1 and P2 into the input to the strategy network. As an example, P1's input at the very last decision step a vector $[x_1, y_1, v_{x_1}, v_{y_1}, x_2, y_2, v_{x_2}, v_{y_2}, \mathtt{type}, u_{1_x}, u_{1_y}, d_{1_x}, d_{1_y}, u_{2_x}, u_{2_y}, d_{2_x}, d_{2_y}, u_{3_x}, u_{3_y}, d_{3_x}, d_{3_y}] \in \mathbb{R}^{21}$, where $u_j$ and $d_j$ represent actions of P1 and P2, respectively, at $j^{th}$ decision point. P2's input, on the other hand, is the same without the $\mathtt{type}$ information making it a vector in $\mathbb{R}^{20}$.

## H.5. Sample Trajectories

### H.5.1. CAMS VS DEEPCFR VS JPSPG

Here we present sample trajectories for three different initial states for each P1 type. The policies learned by CAMS results in trajectories that are significantly close to the ground truth than the other two algorithms.

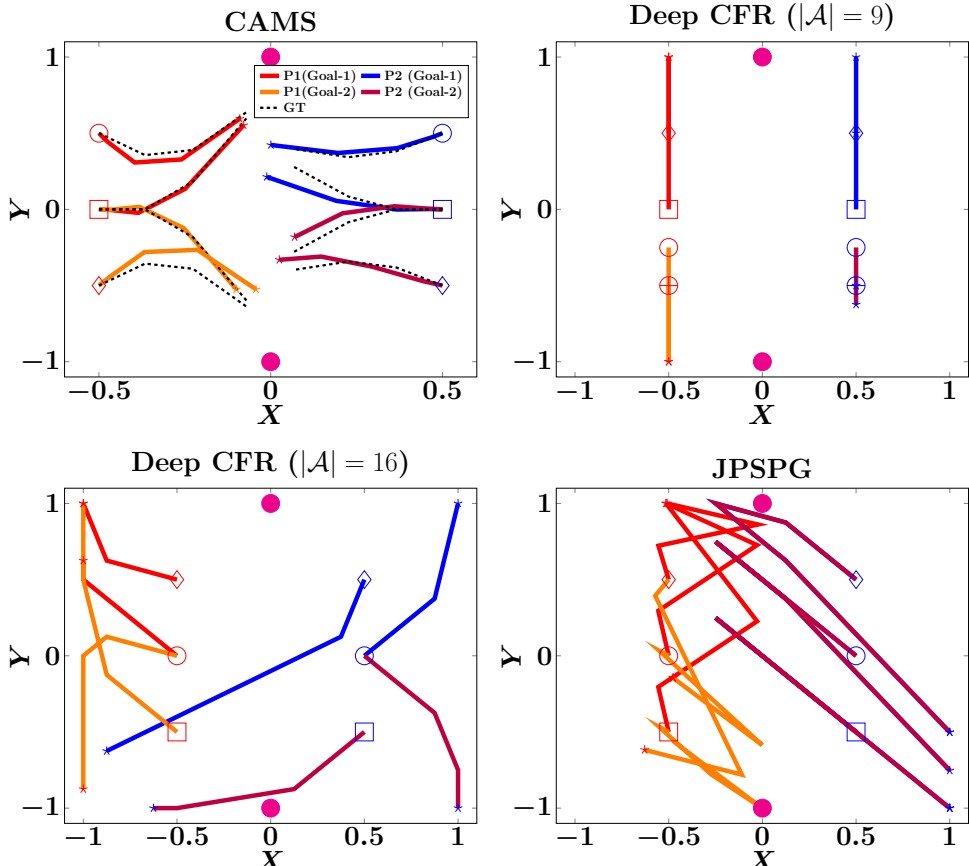

*Figure 9.* Trajectories generated using CAMS (primal game), DeepCFR, and JPSPG. The initial position pairs are marked with same marker and the final with star. The trajectories from CAMS are close to the ground-truth while those from DeepCFR and JPSPG are not.

### H.6. Value Network Training Details

**Data Sampling:** At each time-step, we first collect training data by solving the optimization problem ($P_1$ or $P_2$). Positions are sampled uniformly from [-1, 1] and velocities from $[-\bar{v}_t, \bar{v}_t]$ computed as $\bar{v}_t = t \times u_{max}$, where $u_{max}$ is the maximum acceleration. For the unconstrained game, $u_{max} = 12$ for both P1 and P2. For the constrained case, $u_{x_{max}} = 6$, $u_{y_{max}} = 12$ for P1 and $u_{x_{max}} = 6$, $u_{y_{max}} = 4$ for P2. During training, the velocities are normalized between [-1, 1]. The belief $p$ is then sampled uniformly from $[0, 1]$. For the dual value, we first determine the upper and lower bounds of $\hat{p}$ by computing the sub-gradient $\partial_p V(t_0, \cdot, \cdot)$ and then sample uniformly from $[\hat{p}^-, \hat{p}^+]$.

**Training:** We briefly discuss the training procedure of the value networks. As mentioned in the main paper, both the primal and the dual value functions are convex with respect to $p$ and $\hat{p}$ respectively. As a result, we use Input Convex Neural Networks (ICNN) (Amos et al., 2017) as the neural network architecture. Starting from $T - \tau$, solutions of the optimization problem $P_1$ for sampled $(X, p)$ is saved and the convex value network is fit to the saved training data. The model parameters are saved and are then used in the optimization step at $T - 2\tau$. This is repeated until the value function at $t = 0$ is fit. The inputs to the primal value network are the joint states containing position and velocities of the players $X$ and the belief $p$.

The process for training the dual value is similar to that of the primal value training. The inputs to the dual value network are the joint states containing position and velocities of the players $X$ and the dual variable $\hat{p}$.

## I. Details on Constrained Game

Here, we briefly explain the optimization problem for the constrained game. Formally, given the states $x_1, x_2$ of the players P1 and P2, the constraint is given by the function $c(x_1, x_2) = r - ||(p_{x_1}, p_{y_1}) - (p_{x_2}, p_{y_2})||_2$. P1 must always maintain a radial distance of $r$ from P2, else P1 receives a $+\infty$ penalty and P2 receives a reward of $-\infty$ (both want to minimize their costs).

We follow the method outlined in (Ghimire et al., 2024), and train a separate value function model $\mathcal{F} : t \times \mathcal{X} \to \mathbb{R}$, that classifies the state-space into safe ($\Omega_t$) and unsafe states. Safe states are those initial states from which P1 can avoid collision with P2, whereas unsafe states are those initial states from which it is impossible for P1 to avoid collision. The sub-zero level set of $\mathcal{F}$ correspond to the unsafe states.

With $\mathcal{F}$ available, we can query it to check if the resulting states $x^k = \text{ODE}(x, \tau, u^k, v^k; f)$ in Eq. $P_1$ are unsafe. If so, a high penalty is added (subtracted in the case of dual game), otherwise 0. Formally, at some time-step $t$, initial state $x \in \Omega_t$, $p$ (resp. $\hat{p}$), we solve the following optimization problem for the constrained primal ($V$) (resp. dual ($V^*$)) value:

$$\min_{\{u^k\}, \{\alpha_{ki}\}} \max_{\{v^k\}} \sum_{k=1}^{I} \lambda^k \left( V(t + \tau, x^k, p^k) + \tau \mathbb{E}_{i \sim p^k}[l_i(u^k, v^k)] \right) + \gamma \cdot \texttt{relu}(-\mathcal{F}(t + \tau, x^k))$$

$$\text{s.t.} \quad u^k \in \mathcal{U}, \quad x^k = \text{ODE}(x, \tau, u^k, v^k; f), \quad v^k \in \mathcal{V}, \quad \alpha_{ki} \in [0, 1], \tag{63}$$

$$\sum_{k=1}^{I} \alpha_{ki} = 1, \quad \lambda^k = \sum_{i=1}^{I} \alpha_{ki} p[i], \quad p^k[i] = \frac{\alpha_{ki} p[i]}{\lambda^k}, \quad \forall i, k \in [I].$$

$$\min_{\{v^k\}, \{\lambda^k\}, \{\hat{p}^k\}} \max_{\{u^k\}} \sum_{k=1}^{I+1} \lambda^k \left( V^*(t + \tau, x^k, \hat{p}^k - \tau l(u^k, v^k)) \right) - \gamma \cdot \texttt{relu}(-\mathcal{F}(t + \tau, x^k))$$

$$\text{s.t.} \quad u^k \in \mathcal{U}, \quad v^k \in \mathcal{V}, \quad x^k = \text{ODE}(x, \tau, u^k, v^k; f), \quad \lambda^k \in [0, 1], \tag{64}$$

$$\sum_{k=1}^{I+1} \lambda^k \hat{p}^k = \hat{p}, \quad \sum_{k=1}^{I+1} \lambda^k = 1, \quad k \in [I+1].$$

where $\gamma$ is a scaling factor.

# J. Multigrid Algorithms and Trajectories

Here we present the two multigrid algorithms: 2-level and $n$-level. $n$-level multigrid algorithm can be used to approximate value functions for larger time-steps (or finer discretizations). $n$-level multigrid algorithm is similar to 2-level, with the difference being that the fine grid is coarsened $n$ times, such that the coarsest grid is of at least size 2. We solve the 16-stage game using 4-level multigrid.

---

**Algorithm 2** Two-grid Value Approximation (Multigrid)

---

**Input:** Coarse time-steps $\mathcal{T}^{2l}$, fine time-steps $\mathcal{T}^l$, coarse minimax solver $\mathbb{O}^{2l}$, fine minimax solver $\mathbb{O}^l$, number of data
      points $N$ to sample, restriction operator $\mathcal{R}$, prolongation Operator $\mathcal{P}$
**Initialize:** Initialize fine grid value networks $\hat{V}^l_t \ \forall t \in \mathcal{T}_l$, policy set $\Pi = \varnothing$
**while** *resource not exhausted or until convergence* **do**
    Initialize: $R^l = \varnothing, E^{2l} = \varnothing, \mathcal{S} = \varnothing$,
     Coarse grid correction networks $\varepsilon^{2l}_t \ \forall t \in \mathcal{T}^{2l}$
    $\mathcal{S}[t] \leftarrow$ sample $N$ data points $(t, x, p), \forall t \in \mathcal{T}^l$
    `// Smoothing Step: Few iterations`
    $\texttt{target}, \pi_t \leftarrow \mathbb{O}^l(t + l, \cdot, \cdot, \hat{V}^l_{t+l})$ (init. w/ $\pi_t$ if $\Pi \neq \varnothing$) Store residuals: $r^l_t = (\hat{V}^l_t - \texttt{target})$ in $R^l$ and policies $\pi_t$
    in $\Pi$
    `// Restriction and coarse-solve`
    **for** $t \leftarrow T - 2l$ **to** $0$ **do**
        $e^{2l}_t = \mathbb{O}^{2l}(\mathcal{R}\hat{V}^l_{t+2l} + \varepsilon^{2l}_{t+2l}) - \mathbb{O}^{2l}(\mathcal{R}V^l_{t+2l}) - \mathcal{R}r^l_t$ ;                 `// ` $e^{2l}_T = 0, \ \varepsilon^{2l}_T = \emptyset$
        Store $e^{2l}_t$ in $E^{2l}$
        Fit the correction network $\varepsilon^{2l}_t$ to $e^{2l}_t$
    `// Prolongation`
    **foreach** $t \in \mathcal{T}^l$ **do**
        $e^l_t = \mathcal{P}e^{2l}_t$
        Fit $\hat{V}^l_t$ to $\hat{V}^l_t + e^l_t$
    $\texttt{target}, \pi_t \leftarrow \mathbb{O}^l(t + h, \cdot, \cdot, \hat{V}^l_{t+l})$ (init. with $\pi_t$)
    Fit $\hat{V}^l_t$ to $\texttt{target}$ and replace $\pi_t$ in $\Pi$ ;                              `// Post Smoothing`

---

**Multigrid Trajectories Comparison (Primal Game).** In Fig.10 we compare the trajectories using learned value function via the multigrid approach with the ground truth. The trajectories closely resemble the ground truth with Player 1 successfully able to conceal his true type. Unlike in Fig. 10, where P2's trajectories are due to best response action to P1's action, in Fig. 11, we plot the resulting trajectories when P2 plays the dual game.

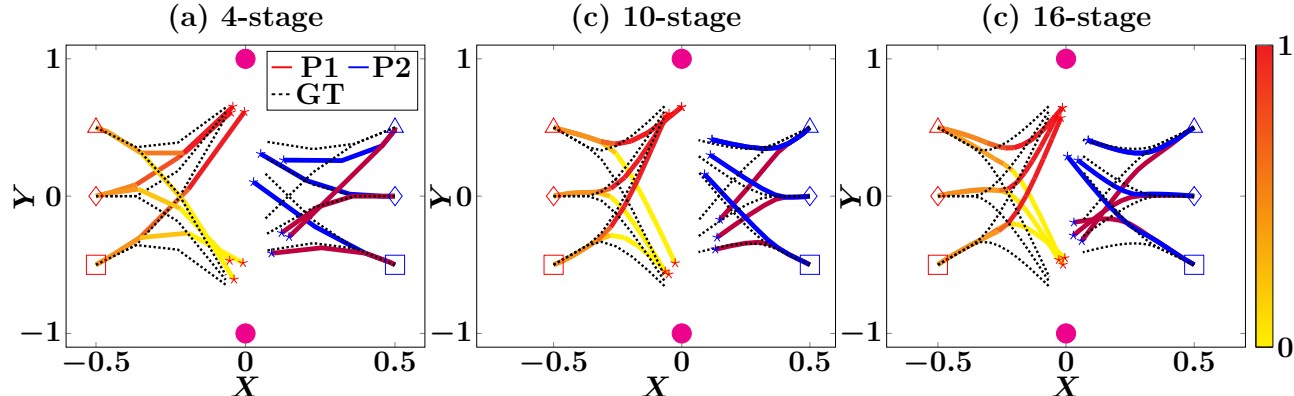

*Figure 10.* Comparison of trajectories generated using value learned via multigrid method vs the ground truth.

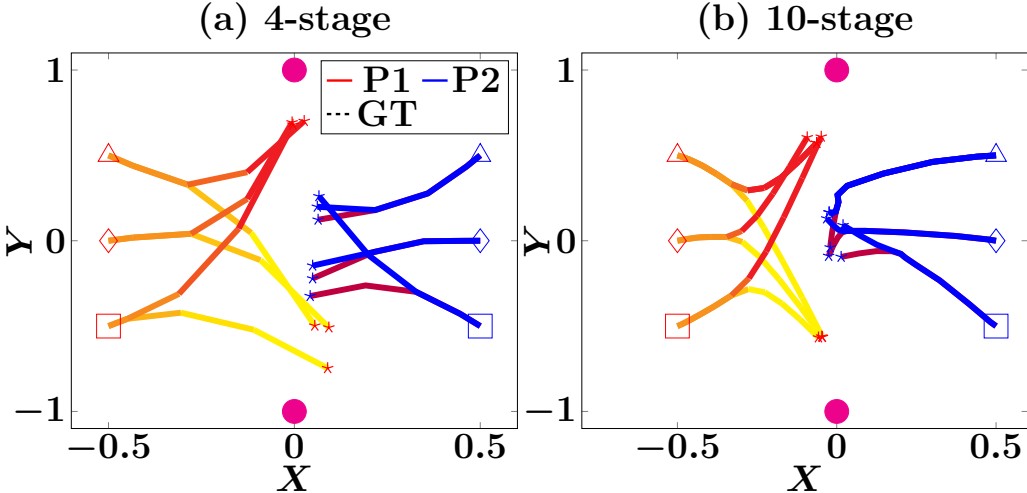

*Figure 11.* Trajectories when P1 and P2 play their respective primal and dual game. P1's actions are a result of the primal value function whereas P2's actions are a result of the dual value function. Both primal and dual values are learned using multigrid approach.

---

**Algorithm 3** $n$-Level Multigrid for Value Approximation

---

**Input:** $k_{\max}, k_{\min}, \mathbb{O}$ (minimax solver), $T$ (time horizon), $N$ (number of data points), $\mathcal{R}$ (restriction operator), $\mathcal{P}$ (prolongation operator)

**Initialize:** $\mathcal{T}^l \leftarrow [0, l, 2l, \ldots, T - l], \forall l \in \{2^{-k_{\max}}, \ldots, 2^{-k_{\min}}\}, L \leftarrow 2^{-k_{\min}}$

**Initialize:** Value networks $\hat{V}_t^l, \forall t \in \mathcal{T}^l, \forall l \in \{2^{-k_{\max}}, \ldots, 2^{-k_{\min}}\}$, policy set $\Pi \leftarrow \varnothing$

**while** *resources not exhausted or until convergence* **do**

    $R \leftarrow \varnothing, E^L \leftarrow \varnothing, \mathcal{S} \leftarrow \varnothing$

    Initialize coarsest-grid correction networks $\varepsilon_t^L, \forall t \in \mathcal{T}^L$

    $\mathcal{S}[t] \leftarrow$ sample $N$ $(t, x, p), \forall t \in \mathcal{T}^{k_{\max}}$

    // down-cycle

    **for** $k \leftarrow k_{\max}$ ***down to*** $k_{\min} + 1$ **do**

        Compute target via $\mathbb{O}^k$ (init. with $\pi_t$ if $\Pi[k] \neq \varnothing$), and store updated policies $\pi_t$ in $\Pi[k], \forall t \in \mathcal{T}^k$

        Compute residuals $r^k[t], \forall t \in \mathcal{T}^k$

        **if** $k \neq k_{\max}$ **then**

            $r_t^k \leftarrow r_t^k + \mathcal{R} r_t^{k+1}, \forall t \in \mathcal{T}^k$

        Store $r_t^k$ in $R[k]$

    **for** $t \leftarrow T - L$ **to** $0$ **do**

        // coarse-solve backwards in time

        $e_t^L \leftarrow \mathbb{O}^L(\mathcal{R}\hat{V}_{t+L}^l + \varepsilon_{t+L}^L) - \mathbb{O}^L(\mathcal{R}\hat{V}_{t+L}^l) - \mathcal{R} r_t^{k_{\min}+1}$ ;        // $e_T^L = 0, \varepsilon_T^L = \varnothing$

        Store $e_t^L$ in $E^L$

        Fit $\varepsilon_L^t$ to $e_L^t$

    // up-cycle

    **for** $k \leftarrow k_{\min} + 1$ ***to*** $k_{\max}$ **do**

        $e_t^k \leftarrow \mathcal{P}(e_t^{k-1}), \forall t \in \mathcal{T}^k$

        Update $\hat{V}_t^k \leftarrow \hat{V}_t^k + e_t^k$

    // post smoothing (for all t's and l's)

    target, $\pi_t \leftarrow \mathbb{O}^l(\hat{V}_{t+l}^l)$ (initialized with $\pi_t$)

    Fit $\hat{V}_t^l$ to target and replace $\pi_t$ in $\Pi[l]$

## K. High Dimensional Hexner's Game

**3D Hexner's game.** To demonstrate the scalability of CAMS, we solve a 3D Hexner's game where the joint action space is now 6D. Accordingly, the state space becomes 12D and the value becomes 13D. Resultant trajectories are visualized in Fig. 12. Similar to the 2D case, P1 learns to correctly conceal his target until some critical time.

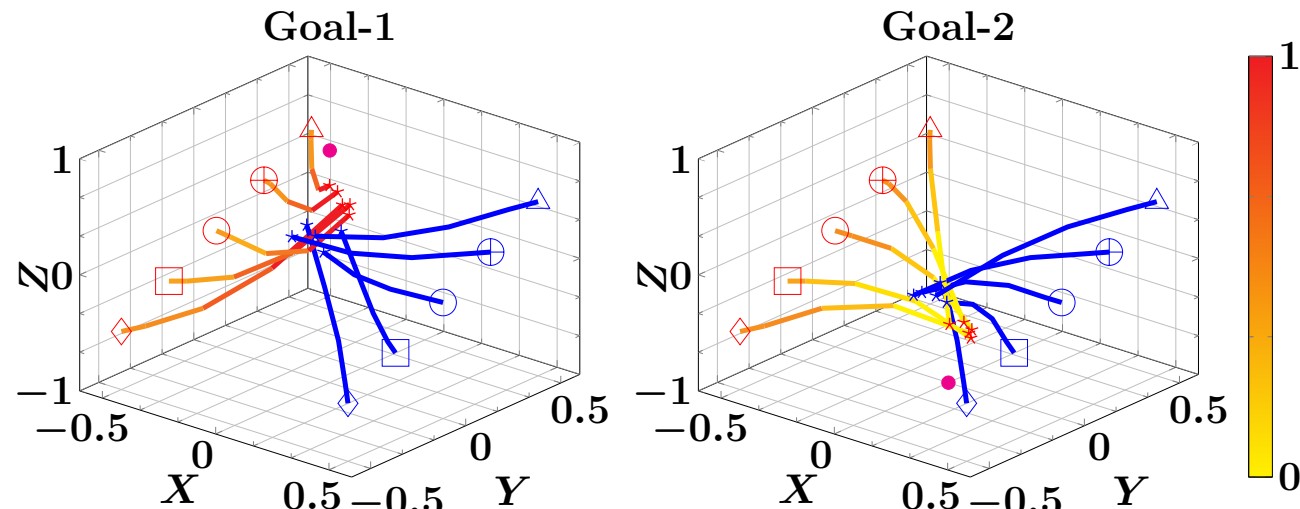

*Figure 12.* 3D Hexner's Game. Color shades indicate the current public belief.

