# OpenReview forum: "A Scalable Solver for 2p0s Differential Games with One-Sided Payoff Information and Continuous Actions, States, and Time"
_ICML.cc/2025/Conference — Submitted to ICML 2025_

### Official Review · Reviewer_ZFnz · 2025-03-07

**Overall Recommendation:** 2

**Summary:**

This paper investigates how to solve 2-player zero-sum EFGs with continuous state and action set, when one of the players has perfect information. They also justify the result with experiments.

**Claims And Evidence:**

It is not clear to me since the proof sketch of the main result Theorem 4.1 is very short and confusing. Moreover, I did not find the definition of $I+1$-atomic in the paper.

**Essential References Not Discussed:**

No.

**Experimental Designs Or Analyses:**

I checked section 6 and the results are reasonable.

**Methods And Evaluation Criteria:**

The Hexner's games seem to be an appropriate benchmark game for this paper.

**Other Comments Or Suggestions:**

Aligning the notations in the paper with EFG literature, such as MMD [2], would be helpful.

**Other Strengths And Weaknesses:**

It would be helpful to evaluate the algorithm in more benchmark games.

**Questions For Authors:**

In assumption A.5, it says both players need to know the Nash equilibrium. I'm wondering if this is a typo since the paper's motivation is to solve the Nash equilibrium to my understanding.

**Relation To Broader Scientific Literature:**

The paper extends the previous results on discrete two-player zero-sum EFGs to EFGs with continuous state and action sets. However, the algorithm is restricted to EFGs with one of the players having perfect information.

**Theoretical Claims:**

I think the proof sketch in the main paper is too short to convince me of its correctness.

---

> ### Author Rebuttal · Authors · 2025-03-31
>
> **[Q3.1]** It is not clear to me since the proof sketch of the main result Theorem 4.1 is very short and confusing. Moreover, I did not find the definition of I+1-atomic in the paper.
>
> **[A3.1]** Here is an extended explanation of Thm. 4.1 which we prove in detail in App. B (and more explanation in [**[A1.2]** for R1](https://openreview.net/forum?id=iDnwpbn20h&noteId=ipvrpvuKt6)). We first explain in terms of the primal game from where P1 finds his strategy. The key idea is that once we reformulate the primal game (let's call it $G_0$) so that P2 plays best responses to P1’s actions to be taken, we can show that the value of this reformulated game ($G_1$) at any belief state $p$ can be written as the convex envelope of an alternative value for a game where both players only play pure strategies ($G_2$). $G_2$ (called the non-revealing game in the paper) is complete-information since its minimax objective is defined as the expected payoff over types, and its value existence is guaranteed by Isaacs' condition. Since a convex envelope on an $I$-dimensional simplex is defined by at most $I$ vertices, P1's strategy in $G_1$ is I-atomic, i.e., the probability distribution of the mixed strategy is concentrated at $I$ actions in $\mathcal{U}$. This idea is explained in “A visual example” on page 4 and visualized in Fig. 2.  The same applies to the dual game for P2 with a small difference: Since the dual game uses a dual state $\hat{p} \in \mathbb{R}^I$ instead of the belief state $p \in \Delta(I)$, P2’s strategy becomes $I+1$-atomic. We are **more than willing** to address further questions from the reviewer regarding Thm 4.1.
>
> **[Q3.2]** The paper extends the previous results on discrete two-player zero-sum EFGs to EFGs with continuous state and action sets. However, the algorithm is restricted to EFGs with one of the players having perfect information.
>
> **[A3.2]** This is correct. We chose this scope because (1) there is a strong demand for computing strategies for 2p0s differential games in real-world applications (e.g., missile defense) that is not being met by SOTA EFG solvers due to their lack of scalability for continuous action spaces [1-4]; and (2) there is currently no algorithm for computing Nash of 2p0s differential games with incomplete information on **both sides** [5; Cardaliaguet, personal communication, Feb 13, 2025]. Even within the scope of one-sided incomplete-information games, there are few studies on explaining the atomic structure of their Nash or the scalability of computing Nash for this type of games, which are the gaps we attempt to fill with this paper. From a practical perspective, we argue that modeling the information as one-sided and deriving a conservative strategy is necessary for the defender (the uninformed player P2) in any risk-sensitive settings, e.g., missile defense.
>
> **[Q3.3]** Aligning the notations in the paper with EFG literature such as MMD would be helpful.
>
> **[A3.3]** We used a different notation to facilitate the discussion about challenges with continuous actions/time and their solutions. That said, there is indeed a connection between behavioral-form MMD and DS-GDA which we used in this paper to solve the reformulated games following Thm. 4.1: While MMD constrains its policy updates with respect to the previous iterate and the magnet, DS-GDA constrains with respect to the moving average. Theoretically, MMD has convergence guarantees only when the objective function is convex-concave (i.e., for sequence-form game formulation) and when the Nash is interior [6]. DS-GDA, on the other hand, has guaranteed convergence even when the objective is nonconvex-nonconcave (i.e., behavioral-form formulation) and when Nash is on the boundary [7].
>
> **[Q3.4]** In A.5, it says both players know the Nash equilibrium. Is this a typo since the paper's motivation is to solve the Nash equilibrium to my understanding?
>
> **[A3.4]** Assuming that both players know the Nash equilibrium does not indicate that we (who study the game) know the Nash equilibrium. In fact, this assumption, i.e., both players know and thus play their Nash strategy, is standard in game theory settings.
>
> [1] Martin, C., T. Sandholm. "Finding mixed-strategy equilibria of continuous-action games without gradients using randomized policy networks." arXiv:2211.15936
>
> [2] Martin, C., T. Sandholm. "Joint-perturbation simultaneous pseudo-gradient." arXiv:2408.09306
>
> [3] Brown, N., et al. "Deep counterfactual regret minimization." ICML 2019
>
> [4] Tammelin, O.. "Solving large imperfect information games using CFR+." arXiv:1407.5042
>
> [5] Cardaliaguet, P. "Differential games with asymmetric information." SIAM journal on Control and Optimization 46.3 (2007): 816-838
>
> [6] Sokota, S., et al. "A Unified Approach to Reinforcement Learning, Quantal Response Equilibria, and Two-Player Zero-Sum Games." ICLR 2023
>
> [7] Zheng, T., et al. "Universal gradient descent ascent method for nonconvex-nonconcave minimax optimization." NeurIPS 2023

---

> > ### Comment · Reviewer_ZFnz · 2025-04-03
> >
> > Thank you for your response! Could you elaborate on how you used assumption A5 in the paper and provide some reference that also assumed it?

---

> > > ### Author Response · Authors · 2025-04-04
> > >
> > > Certainly, let us reiterate the assumption.
> > >
> > > **A5**: Both players have full knowledge about (1) $f$ (dynamics), (2) $l, g$ (payoffs), (3) $p_0$ (prior public belief), (4) and the Nash equilibrium. (5) Control inputs, and (6) states are fully observable and we assume (7) perfect recall.
> > >
> > > **How we used A5**
> > > Together with A1-4, A5 (except (4)) forms the sufficient conditions for the presented game to have a value, and therefore a Nash equilibrium (NE) [1]. In general, (4) is required for players in IIEFGs to update their beliefs about each other (e.g., to update P1’s belief about P2’s private cards in a Poker game, P1 needs to know P2’s NE). We do acknowledge that this assumption is stronger than necessary in games with one-sided information: while the uninformed player (P2) needs to know the NE of the informed player (P1), P1 does not need to know P2’s NE since P2 has no private info (this is first discussed in [2]).
> > >
> > > **References that assumed A5**
> > > - **From the side of EFGs**: An important set of IIEFG studies assume A5 (e.g., [3-6]). Taking Poker as an example, game dynamics and payoffs are pre-defined; the true payoff type is defined by the private cards and therefore $p_0$ can be pre-calculated; all player actions and states (chips and public cards) are observable; and all previous actions of the game are assumed to be memorized by players. We do note that there is a broader set of IIEFGs and POMGs that assume partial observability, in which case some dynamically changing system states, rather than payoff types, are private, and thus (6) is not assumed [4, 7]. Our paper does not consider partial observability.
> > >
> > > - **From the side of differential games**: Almost all optimal control and differential game papers assume A5 (see description of Thm 1.3 of [9], page 11 line 4 of [11], and [10]), although the majority of them (refs in [12]-[14] for example) assume complete information, in which case (3) does not apply.
> > >
> > > [1]Cardaliaguet, Pierre. "Differential games with asymmetric information." SIAM journal on Control and Optimization 46.3 (2007): 816-838.
> > > [2]De Meyer, Bernard. "Repeated games, duality and the central limit theorem." Mathematics of Operations Research 21, no. 1 (1996): 237-251.
> > > [3] Zinkevich, Martin, et al. "Regret minimization in games with incomplete information." Advances in neural information processing systems 20 (2007).
> > > [4] Schmid, Martin, et al. "Student of Games: A unified learning algorithm for both perfect and imperfect information games." arXiv preprint arXiv:2112.03178 (2021).
> > > [5] Perolat, Julien, et al. "From poincaré recurrence to convergence in imperfect information games: Finding equilibrium via regularization." International Conference on Machine Learning. PMLR, 2021.
> > > [6] Hennes, Daniel, et al. "Neural replicator dynamics: Multiagent learning via hedging policy gradients." Proceedings of the 19th international conference on autonomous agents and multiagent systems. 2020.
> > > [7] Brown, Noam, et al. "Combining deep reinforcement learning and search for imperfect-information games." Advances in neural information processing systems 33 (2020): 17057-17069.
> > > [8] Berkovitz, Leonard D. "Two person zero sum differential games: An overview." The Theory and Application of Differential Games: Proceedings of the NATO Advanced Study Institute held at the University of Warwick, Coventry, England, 27 August–6 September, 1974. Dordrecht: Springer Netherlands, 1975.
> > > [9] Cardaliaguet, Pierre. "Information issues in differential game theory." ESAIM: Proceedings. Vol. 35. EDP Sciences, 2012.
> > > [10] MICHAEL, D. DIFFERENTIAL GAMES: A CRITICAL VIEW.
> > > [11] Asher, Robert Bernard. DIFFERENTIAL GAMES WITH SYSTEM UNCERTAINTY AND IMPERFECT INFORMATION. Oklahoma State University, 1974.
> > > [12] Bansal, Somil, Mo Chen, Sylvia Herbert, and Claire J. Tomlin. "Hamilton-jacobi reachability: A brief overview and recent advances." In 2017 IEEE 56th Annual Conference on Decision and Control (CDC), pp. 2242-2253. IEEE, 2017.
> > > [13] Bansal, Somil, and Claire J. Tomlin. "Deepreach: A deep learning approach to high-dimensional reachability." In 2021 IEEE International Conference on Robotics and Automation (ICRA), pp. 1817-1824. IEEE, 2021.
> > > [14] Gammoudi, Nidhal, and Hasnaa Zidani. "A differential game control problem with state constraints." Mathematical Control and Related Fields (2022).

---

### Official Review · Reviewer_tRrX · 2025-03-12

**Overall Recommendation:** 3

**Summary:**

This work tackles an imperfect-information extensive-form games (IIEFGs) often struggle with continuous action and state spaces and continuous time. This paper addresses the scalability challenges for 2p0s game with one-sided information. By showing an atomicity property and using the same the authors show that the complexity of computing the Nash equilibrium for these games is not related to the
action space size; and they show a multi-grid approach of solving faster.

**Claims And Evidence:**

The approach is convincing, but could be written more clearly. See other Strengths And Weaknesses.

**Essential References Not Discussed:**

Seems fine.

**Experimental Designs Or Analyses:**

Experiments are fine.

**Methods And Evaluation Criteria:**

Yes, they do.

**Other Comments Or Suggestions:**

N/A

**Other Strengths And Weaknesses:**

This work is not in my area, particularly the numerical solution of PDE.
From what I gather, inspired by structural results from prior work, the authors study computational issues in implementing these.

On line 250, second column what is d_u - where is this defined? And, this claim made: "computational complexity of these problems are no longer related to the size of the action spaces" - please describe in detail, mainly the proof sketch after the minmax formulation was not clear to me. It might also help to explain FAS, especially for readers like me who are not familiar with it (some past results could be compressed)

**Questions For Authors:**

Respond to points in other Strengths And Weaknesses

**Relation To Broader Scientific Literature:**

The contributions as claimed are important.

**Theoretical Claims:**

Not checked in the supplementary, all proofs are in supplementary which is not very helpful. See other Strengths And Weaknesses.

---

> ### Author Rebuttal · Authors · 2025-03-31
>
> **[Q2.0]** From what I gather, inspired by structural results from prior work, the authors study computational issues in implementing these.
>
> **[A2.0]** Thank you. We would like to highlight that to the authors’ best knowledge, this is the first paper that concretely explained the atomic nature of Nash equilibrium for 2p0s differential games with one-sided information. Therefore we humbly claim both theoretical and computational contributions.
>
> **[Q2.1]** On line 250, second column what is d_u?
>
> **[A2.1]** $d_u$ is the dimension of action $u$.
>
> **[Q2.2]** "computational complexity of these problems are no longer related to the size of the action spaces" - please describe in detail, mainly the proof sketch after the minmax formulation was not clear to me.
>
> **[A2.2]** Please refer to our response [**[A3.1]** to **[Q3.1]** raised by R3 (ZFnz)](https://openreview.net/forum?id=iDnwpbn20h&noteId=UhBmjpGVXh)
>
> **[Q2.3]** It might also help to explain FAS, especially for readers like me who are not familiar with it (some past results could be compressed)
>
> **[A2.3]** We will describe FAS in revision to convey the following main idea:
>
> Multigrid methods were first developed to accelerate linear solvers for
> $Ax = b$. Let the solution approximation be $y$, error be $e = x - y$, and residual be $r = b - Ay$. Starting from an initial $y$, a multigrid V-cycle transfers $r$ from the fine grid to the coarse grid, where $e$ is computed from $Ae = r$ to resolve low-frequency residual and interpolated back to the fine grid to refine $y$. FAS is an extension of this idea to accelerate nonlinear solvers for $f(x)=b$, where $r$ is resolved by solving a nonlinear problem "fully" to obtain an exact solution for $x$ on the coarse grid. We refer readers to [3] for a comprehensive review of FAS.
>
> For both linear and nonlinear problems, multigrid (and FAS) leverages multiple mesh levels to resolve residuals associated with different error frequencies [1]. While FAS broadens the scope of multigrid to nonlinear systems, its practical utility depends on provable convergence guarantees. These guarantees typically include showing that each iteration of the multigrid cycle (e.g., a V-cycle) contracts the error by a factor $\gamma$ (the convergence rate) that is **independent of the mesh size**. Mathematically, this is expressed as $ ||f(y_{k+1}) - x|| \le \gamma ||f(y_k) - x|| $, where $y_k$ is the approximation at the $k$-th iteration, and $x$ is the true solution. The existence of a contraction factor $0 < \gamma < 1$, independent of the mesh size, is established in [7] for linear PDEs and in [8] for certain classes of nonlinear elliptic PDEs. Although a theoretical convergence rate for Hamilton-Jacobi-Isaacs (HJI) PDEs has not yet been established, [4] provides empirical evidence indicating that the convergence rate of FAS in an optimal control problem (governed by an Hamilton-Jacobi-Bellman equation) is bounded by 0.1. To the best of our knowledge, no analogous results are currently available for HJI PDEs.
>
> [1] Olson, Luke. Multigrid Methods. https://lukeo.cs.illinois.edu/files/2015_Ol_encmg.pdf.
> [2] Vegt van der, Jaap. Introduction to Multigrid Methods. https://faculty.ustc.edu.cn/_tsf/00/2A/FjIjI3fAJjqa.pdf.
> [3] Henson, Van. "Multigrid methods nonlinear problems: an overview." Computational imaging 5016 (2003): 36-48.
> [4] Han, Dong, and Justin WL Wan. "Multigrid Methods for Second Order Hamilton--Jacobi--Bellman and Hamilton--Jacobi--Bellman--Isaacs Equations." SIAM Journal on Scientific Computing 35.5 (2013): S323-S344.
> [5] Akian, Marianne, and Sylvie Detournay. "Multigrid methods for two‐player zero‐sum stochastic games." Numerical Linear Algebra with Applications 19.2 (2012): 313-342.
> [6]Akian, Marianne, P. Séquier, and Agnks Sulem. "A finite horizon multidimensional portfolio selection problem with singular transactions." Proceedings of 1995 34th IEEE Conference on Decision and Control. Vol. 3. IEEE, 1995.
> [7] Braess, D., and W. Hackbusch. “A New Convergence Proof for the Multigrid Method Including the V-Cycle.” SIAM Journal on Numerical Analysis, vol. 20, no. 5, 1983, pp. 967–75. JSTOR, http://www.jstor.org/stable/2157109. Accessed 31 Mar. 2025.
> [8] Reusken, A. (1988). Convergence of the multilevel Full Approximation Scheme including the V-cycle. Numerische Mathematik, 53(6), 663–686. doi:10.1007/bf01397135.

---

### Official Review · Reviewer_JAWG · 2025-03-15

**Overall Recommendation:** 3

**Summary:**

This paper focuses on two-player zero-sum differential games with continuous actions, states, and time. The author first proves that equilibria of these games can be computed by solving a dynamic programming problem with a discrete-time approximation. It is then shown that the complexity of this dynamic programming is independent of the size of the action space. Finally, a multigrid approach is proposed, and its performance is empirically evaluated through experiments.

**Claims And Evidence:**

The claims are supported by theoretical results and numerical experiments.

**Essential References Not Discussed:**

While I am not deeply familiar with differential games, I wonder if there are other studies on games with continuous action spaces beyond (Martin & Sandholm, 2023; 2024).

**Experimental Designs Or Analyses:**

The paper presents numerical experiments on Hexner's game to validate the proposed solver and compares its performance with existing state-of-the-art solvers. However, there seems to be a discrepancy regarding the reported computational times. In Section 6.2, it is stated that "The wall-time costs for game solving are 17 hours using CAMS," while Figure 3(a) indicates much lower wall times. It would be helpful if the author could clarify what the y-axis in Figure 3(a) represents.

**Methods And Evaluation Criteria:**

Leveraging the fact that the computational complexity of solving the dynamic programming is independent of the action space size is intriguing. However, the benefit of the multigrid approach in addressing the fine time-discretization issue is not fully clear. The paper would benefit from a more detailed explanation of why the multigrid method is able to resolve this issue.

**Other Comments Or Suggestions:**

In Section 2, it is stated that “Since any 2p0s IIEFG with finite action sets has a normal-form formulation, a unique Nash equilibrium always exists in the space of mixed strategies.” However, two-player zero-sum games can have multiple Nash equilibria in terms of strategy profiles, although the value of the game is unique. What did you mean by “unique” in this context?

**Other Strengths And Weaknesses:**

I don't have anything in particular to comment here.

**Questions For Authors:**

I don't have anything in particular to comment here.

**Relation To Broader Scientific Literature:**

Utilizing the structure of differential games to compute equilibria is a novel contribution.

**Theoretical Claims:**

I am particularly interested in Theorem 4.1. After reviewing its proof in Appendix B, I am still unclear on why the optimal solution $v^{\ast}$ in the RHS of Eq. (23) is $I$-atomic. Since this claim appears to be a key element of Theorem 4.1, I would like to know more about this.

Furthermore, I am not sure what Theorem 3.1 intuitively indicates. While the author provided some intuition behind the theorem (in line 190-192), it would be better to provide a more detailed explanation.

---

> ### Author Rebuttal · Authors · 2025-03-31
>
> **[Q1.1]**  The benefit of multigrid in addressing the fine time-discretization issue is not fully clear.
>
> **[A1.1]** Please refer to our response [**[A2.3]** to **[Q2.3]** raised by R2 (tRrx)](https://openreview.net/forum?id=iDnwpbn20h&noteId=vlkPNnyOPK)
>
> **[Q1.2]** Why is the optimal solution in the RHS of Eq. 23 I-atomic?
>
> **[A1.2]** Expanding on App. B, below we show that for fixed $(t_0,x_0)$, $V(\cdot) := V_\tau(t_0,x_0,\cdot)$ as defined in Eq. 23 is the convexification of $a(\cdot)$ on $\Delta(I)$. Since convexification on $\Delta(I)$ requires at most $I$ vertices, the optimal solution is I-atomic.
>
> To recall, Eq. 23: $V_\tau(t_0,x_0,p) = \min_\nu \int_{\Delta(I)} a(p')\nu(dp') \quad \text{s.t.}~\int_{\Delta(I)} p'\nu(dp') = p$,
> where $a(p'): \Delta(I) \rightarrow \mathbb{R}$ and $\nu$ is a probability measure defined on $\Delta(I)$.
>
> (1) We show $V$ is convex on $\Delta(I)$: Let probability measures $\nu^i$ for $i \in \{1,2\}$ be the solution to Eq. 23 for $p^i$. For any $\theta \in [0,1]$ and $p^\theta = \theta p^1 + (1-\theta)p^2$, the mixture $\nu^\theta := \theta \nu^1 + (1-\theta) \nu^2$ satisfies $\int_{\Delta(I)} p'\nu(dp') = \theta \int_{\Delta(I)} p'\nu^1(dp') + (1-\theta) \int_{\Delta(I)} p'\nu^2(dp') = p^\theta$ and is a feasible solution to Eq. 23 for $p^\theta$. Therefore $$V(p^\theta) \leq \int_{\Delta(I)} a(p')\nu^\theta(dp') = \theta V(p^1) + (1-\theta) V(p^2).$$
>
> (2) We show $V(p) \leq a(p)$ for any $p \in \Delta(I)$: The Dirac measure $\delta_p$ concentrated at $p$ satisfies $\int_{\Delta(I)} p'\delta_p(dp') = p$. Thus $\delta_p$ is a feasible solution to Eq. 23 and by definition $V(p) \leq \int_{\Delta(I)} \tilde{a}(p')\delta_p(dp') = a(p)$.
>
> (3) We show $V$ is the largest convex minorant of $a$: Let $h$ be any convex function on $\Delta(I)$ such that $h(p)\leq a(p)$ for all $p$. Given $p \in \Delta(I)$, for any probability measure $\nu$ that satisfies $\int_{\Delta(I)} p'\nu(dp') = p$, we have
> $$h(p) = h(\int_{\Delta(I)} p'\nu(dp')) \leq \int_{\Delta(I)} h(p')\nu(dp') \leq \int_{\Delta(I)} a(p')\nu(dp').$$ Since this inequality holds for arbitrary $\nu$, including the optimal ones that define $V(p)$ through Eq. 23, it follows that $h(p) \leq V(p)$. With (1)-(3), $V(\cdot)$ in Eq. 23 is the convexification of $a(\cdot)$.
>
> **[Q1.3]** Intuitive explanation of Thm. 3.1
>
> **[A1.3]** Thm. 3.1 states that the Nash of the reformulated games are $\epsilon$-Nash for the original games, where $\epsilon \rightarrow 0$ as the time interval $\tau \rightarrow 0$. Let us focus on the primal game and explain the two inequalities involved:
>
> (1) $V \leq \max J$: Here $V$ is the value of the original game where players move simultaneously, $\max J$ the value of the reformulated game where P1 moves before P2. (1) holds because P1 has a disadvantage in the reformulation.
>
> (2) $\max J \leq V + M\tau$ says that this disadvantage is bounded by $M\tau$. This uses the fact that during any time interval, the value gap between P2 playing best response vs. any alternative strategy (including the Nash of the original game) is bounded by $\tau^2$ (Eq. 18), thanks to the continuity and boundedness assumptions on the dynamics and payoffs.
>
> **[Q1.4]** A discrepancy in the reported computational times…what does the y-axis in Fig. 3(a) represent?
>
> **[A1.4]**	Fig. 3(a) shows wall times for solving the *1-stage* Hexner’s game. “17-hour” refers to the *4-stage* game. We will better clarify.
>
> **[Q1.5]** Other studies on games with continuous action spaces beyond Martin & Sandholm?
>
> **[A1.5]** Other studies on continuous action games, including those referenced in Martin & Sandholm (2023; 2024), are largely limited to one-shot Bayesian games [1] and auction games [2], which fundamentally differ from differential games with incomplete information. Other algorithms, such as [3], discretize the action space, making them impractical for high-dimensional settings. We chose Martin & Sandholm (2023; 2024) as representative algorithms due to their demonstrated success in solving a continuous action imperfect-information extensive-form game (a continuous action variant of Goofspiel). To the best of our knowledge, SOTA methods for solving incomplete/imperfect-information zero-sum extensive-form games with continuous actions are yet to be established.
>
> **[Q6]** What did you mean by “unique” in this context?
>
> **[A6]** The reviewer is correct: The value is unique, not the strategy.
>
> [1] Zun Li and Michael P Wellman. Evolution strategies for approximate solution of bayesian games. In AAAI Conference on Artificial Intelligence (AAAI), 2021.
> [2] Martin Bichler, Maximilian Fichtl, Stefan Heidekruger, Nils Kohring, and Paul Sutterer. Learning equilibria in symmetric auction games using artificial neural networks. Nature Machine Intelligence, 2021.
> [3] Martin Bichler, Max Fichtl, and Matthias Oberlechner. Computing Bayes–Nash equilibrium strategies in auction games via simultaneous online dual averaging. Operations Research, 2023a.

---

### Decision · Program_Chairs · 2025-05-01

**Decision:**

Reject

**Comment:**

The paper focuses on two-player zero-sum differential games with continuous actions, states. It is shown that approximate equilibria can be found via dynamic programming where its complexity is independent of the size of the action space. Finally, a multigrid approach is proposed and experiments are provided. The reviews were mixed, but none of the positive reviewers showed high excitement for the paper. Also some concerns were raised about the writeup. We recommend weak rejection.